# Recent Developments in the Flame-Retardant System of Epoxy Resin

**DOI:** 10.3390/ma13092145

**Published:** 2020-05-06

**Authors:** Quanyi Liu, Donghui Wang, Zekun Li, Zhifa Li, Xiaoliang Peng, Chuanbang Liu, Yu Zhang, Penglun Zheng

**Affiliations:** 1College of Civil Aviation Safety Engineering, Civil Aviation Flight University of China, Guanghan 618307, China; quanyiliu2005@cafuc.edu.cn (Q.L.); donghuiwang@cafuc.edu.cn (D.W.); lizekun96@cafuc.edu.cn (Z.L.); lizhifa@cafuc.edu.cn (Z.L.); pxl@cafuc.edu.cn (X.P.); liubang@cafuc.edu.cn (C.L.); 2School of Mechanical and Power Engineering, East China University of Science and Technology, 130 Meilong Road, Shanghai 200237, China

**Keywords:** epoxy resin, flame retardant, halogen-free

## Abstract

With the increasing emphasis on environmental protection, the development of flame retardants for epoxy resin (EP) has tended to be non-toxic, efficient, multifunctional and systematic. Currently reported flame retardants have been capable of providing flame retardancy, heat resistance and thermal stability to EP. However, many aspects still need to be further improved. This paper reviews the development of EPs in halogen-free flame retardants, focusing on phosphorus flame retardants, carbon-based materials, silicon flame retardants, inorganic nanofillers, and metal-containing compounds. These flame retardants can be used on their own or in combination to achieve the desired results. The effects of these flame retardants on the thermal stability and flame retardancy of EPs were discussed. Despite the great progress on flame retardants for EP in recent years, further improvement of EP is needed to obtain numerous eco-friendly high-performance materials.

## 1. Introduction

Since the invention of epoxy resin (EP) in the 1930s, significant progress has been achieved in EP which can satisfy different requirements in practical applications. EPs have been widely used as coatings, composite materials, castables, adhesives, molding materials and injection molding materials, owing to the exceptional mechanical properties, dimensional stability, suitable chemical and thermal resistance [1,2,3,4,5,6]. However, EP usually suffers from a high fire risk while used in aircraft, trains and ships owing to its poor flame retardancy [7,8,9]. The combustion of EP causes a high release rate of heat and smoke which could pose serious consequences. Consequently, it is particularly important to modify EP to enhance its flame retardancy without sacrificing other properties [10].

To date, numerous works have been done on flame retardants for EPs [11,12,13,14,15]. The most popular strategies include structural modification or incorporation of flame retardants. Structural modification is to introduce the elements with flame retardant function into the molecular structure of EP. Among these methods, the addition of flame retardants in EP show numerous advantageous properties such as simple processing, low cost, wide source of raw materials and obvious flame-retardant effect. Common flame retardants are based on halogen [16,17], organophosphorus [18,19,20], intumescent effect [21,22,23], silicon-containing compounds [24,25], nanocomposites [26,27] and metal-containing compounds [28,29], etc. However, halogen-containing flame retardants could decompose at high temperatures to produce dioxin, which is a persistent organic pollutant with environmental persistence, bioaccumulation, long-distance migration ability, and high biological toxicity [30]. This poses a serious threat to the ecological environment and human health, and limits the application of halogen flame retardants [31]. Therefore, halogen-free flame retardants have aroused great interest from the perspective of environmental protection due to its non-toxic constituents.

Numerous halogenated flame retardants (such as polybrominated diphenyl ethers) have been banned worldwide due to their growing and continuing threat to the environment and organisms [32]. Many halogen-free flame retardants with flame retardant benefits and less environmental hazards have been used as substitutes for halogen-based flame retardants [33,34]. However, some researchers have found that organophosphate flame retardants were identified as a new type of pollutant because they were detected in environmental and biological samples [35]. The increasing use of flame retardants in worldwide will further increase their environmental concentration. In addition, many replacement chemicals are protected by confidential business information regulations, making it difficult to accurately assess their potential hazards, and the long-term health effects may not be obvious for some time after their introduction to the market. Therefore, their impact on human health and the environment cannot be ignored. Researchers should also pay attention to the cost of ecological health while analyzing the fire safety benefits of using flame retardants.

Currently, research focuses on environmentally friendly flame retardants for EPs include phosphorus-based flame retardants, carbon-based materials, silicon-containing compounds, nanocomposites, and metal-containing compounds. For instance, X. Li [36] et al. used the non-covalent ionic liquid flame retardant functionalized boron nitride nanosheets as nano-additive to produce epoxy-based nanocomposites with unique structure and functionality. The obtained nanocomposite material improves the flame retardancy of EP. Javaid [37] et al. prepared a new type of EP composite by crosslinked diglycidyl ether of braided carbon fibers (CFs) and bisphenol A-epoxy matrix modified by phosphoric acid. The research showed that the fire resistance of this material is outstanding. Y. Guang [38] et al. used N-aminoethyl piperazine and hexachlorocyclotriphosphazonitrile to synthesize a new type of active amine compound containing polyphosphazene (PBFA) and introduced it into EP. It was found that PBFA significantly improved the fire resistance and inhibited smoke generation. The increasing awareness about the flame-retardant EP composites to meet engineering requirements has attracted substantial attention.

This review focuses on recent advances of a few popular flame-retardant systems for EP, including phosphorus flame retardant, carbon-based materials, silicon flame retardants, nanocomposites and metal-containing compounds. The concerns on the practical applications of flame retardants are also involved. It is necessary to ensure that the other properties of the EP are not sacrificed while improving the flame-retardant performance. We believe this review could provide a timely progress report on the flame retardants for EP and benefit junior researchers in this field. 

## 2. Research Progress on Flame Retardants for Epoxy Resin

Currently, some flame retardants used in EPs have been commercialized. Although halogen-based flame retardants have been opposed by many countries, bromine-based flame retardants still play a significant role in the field of flame retardants. Tetrabromobisphenol A is still the single largest flame retardant in terms of current commercial value. It is widely used in printed circuit boards of FR-4 EP, and is priced as a commodity. In addition, some high-load metal hydroxides were used by some Japanese companies. A Hitachi Chemical [39] patent showed that adding 120–160 wt.% Aluminum hydroxide to phenol novolac EP could achieve UL-94 V-0 rating and exceptional soldering temperature resistance. Another formulation of Hitachi Company increased the thermal stability to 288 °C by adding 4 wt.% polysiloxane and 30 wt.% aluminium trihydrate (ATH). Toshiba [40] has developed an adhesive formulation based on bisphenol A and cresol novolac resin. This formulation showed that EP and cresol novolac resin could pass UL-94 V-0 test by adding 4 wt.% red phosphorus (encapsulated in phenolic resin and coated with ATH) and 25 wt.% ATH. The metal salt of dialkyl phosphate was considered to be a flame retardant with broad application prospects due to its good flame retardancy. Clariant [41] described an aluminum salt flame retardant of diethyl phosphite (named Exolit OP 930 and Exolit OP 935), which has broad application prospects in printed wiring board (PWB). In addition, Sanko Corporation [42] of Japan described a commercial flame retardant, a multifunctional compound named HCA-HQ synthesized by the reaction of 9,10-dihydro-9-oxa-10-phosphaphenanthrene 10-oxide (DOPO) and benzoquinone. HCA-HQ could achieve a high flame-retardant effect at a very low phosphorus content (2 wt.%). At present, commercial flame retardants used in EPs have made great progress, but there are still some problems such as large additions resulting in reduced mechanical properties. In order to obtain competitive products in the field of flame-retardant EPs, complex flame retardant system research is required. To date, many potential flame retardants in EP have been explored, including phosphorus-based, carbon-based, silicon-based, nanocomposites and mental-containing compounds. The following sections will summarize the recent progress on these promising flame retardants for EP.

### 2.1. Phosphorus Flame Retardants

Phosphorus-containing flame retardants are important components of halogen-free flame retardants which possesses excellent properties such as low smoke emission, low toxicity, and formation a stable carbonized layer after burning [43,44,45]. The carbonized layer can prevent further pyrolysis of the corresponding polymer, and also inhibit the diffusion of thermally decomposed internal products into the gas phase [46]. Flame retardants based on phosphorus can be divided into inorganic phosphorus and organic phosphorus flame retardants. Inorganic phosphorus flame retardants mainly refer to red phosphorus [47], ammonium polyphosphate (APP) [48], etc. Organophosphorus flame retardants include phosphate esters [49], polyols-containing phosphorus [50], etc. In addition, DOPO and phosphorus-containing synergistic flame-retardant systems have also attracted widespread attention [51,52,53]. Adding a phosphorus flame retardant to the polymer can significantly improve the flame retardancy of the polymer, but it may reduce the thermal stability of the polymer and increase the smoke production during combustion.

#### 2.1.1. Inorganic Phosphorus

Flame retardants based on inorganic phosphorus mainly include red phosphorus, APPs and phosphates. Inorganic phosphorus compounds are halogen-free and non-volatile and have long-lasting flame-retardant effects as well as thermal stability [54]. They are mainly used in polyvinyl chloride (PVC), nylon epoxy, polyester and polyamide, etc. [55]. As an important part of inorganic flame retardants, inorganic phosphorus flame retardants have broad application prospects [17].

APP has attracted attention in recent years because of its high decomposition temperature and good thermal stability. However, a surface modification treatment is required due to its poor compatibility with polymers. A novel APP modified with branched polyethyleneimine (PEI) (named as APP-PEI) was synthesized (Figure 1) [56]. The decoration of PEI for APP can tremendously enhance the compatibility between APP and EP. Experimental results showed that PEI-APP enhanced the flame retardancy of EPs: The limit oxygen index (LOI) values increased up to 29.5% and the vertical burning test achieved UL-94 V-0 rating, total heat release (THR) decreased by 76.1% compared to neat EP. At the same time, exceptional smoke suppression was obtained after curing by PEI-APP: total smoke production (TSP) decreased by 70.5%.

In addition, phosphorus-containing diamine compounds are important components of inorganic phosphorus flame retardants which generally used in the modification of EP. G.-H. Hsiue [57] et al. synthesized two phosphorus-containing diamine compounds (bis(4-aminophenoxy)phenyl phosphine oxide (PA-I) and bis(3-aminophenyl)phenyl phosphine oxide (PA-II)) and utilized them as curing agents of EPs. The research revealed that the residues of combustion higher than 35% in nitrogen and the composites had LOI values of about 35. High char yields and LOI implied that these phosphorus-containing diamines curing agents could improve the flame retardancy of EPs. It has been reported that Cu_2_O could reduce the toxicity of combustion products due to its catalytic reduction coupling effect [58]. Microencapsulated structure can improve the compatibility and water resistance of APP with polymers such as EP. M.-J Chen [59] et al. utilized Cu_2_O and microencapsulated ammonium polyphosphate (MAPP) to modify EPs. The numerical and experimental results showed that the EP with the addition of 18 wt.% MAPP and 2 wt.% Cu_2_O exhibited the best flame retardancy (Table 1). Its LOI value reached up to 35% and UL-94 rating of V-0. In addition, the smoke-reducing and charring rate of the modified EP was also improved. The improvement of flame retardancy of EP could be attributed to the role of MAPP/Cu_2_O in promoting the carbonization and oxidation of CO.

#### 2.1.2. Organic Phosphorus

Organic phosphorus flame retardants are one class of flame retardants with outstanding flame retardants properties for EPs [54]. The organic phosphorus-based flame retardants include phosphonates, phosphate esters, organic phosphorus salts, phosphorus heterocyclic compounds, etc. The main advantages of organic phosphorus flame retardants are low smoke emission, low toxicity and environmental friendliness [46]. At present, organophosphorus flame retardants have received extensive attention. Numerous efforts have been done on the organophosphorus flame retardants [60,61,62].

Hyperbranched polymers have been widely concerned because they possess many important properties of dendrimers that can provide the possibility of further modification. X. Chen [63] et al. synthesized a novel hyperbranched phosphate (HPE) and used as curing agent of EP. Research suggested that HPE used as a curing agent could impart flame retardancy to EP. The LOI value of the material achieved 27.5% when the content of HPE in the modified EP was 33 wt.%. The peak heat release rate (pk-HRR), average of heat release rate (av-HRR), and THR decreased significantly with the increase in HPE content, while char yield increased. In a similar work, the effect of phosphorous-containing hyperbranched polymers on the properties of EPs was investigated by A. Battig [64] et al. They synthesized a series of multifunctional phosphorus-containing hyperbranched polymeric flame retardants (hb-FRs) and added them to EP as additives. The experimental results showed that hb-FRs had a small effect on the LOI value and UL-94 grades of the epoxy composites. The thermal stability of the epoxy composites had been greatly improved. The pk-HRR and THR significantly decreased with increasing hb-FRs content, while residue yield increased. Moreover, they compared the flame retardancy of a series of sulfur-containing and sulfur-free hyperbranched polyphosphate additives [65]. The introduction of sulfur improved the thermal stability of the flame retardant and the activity of the condensed phase. The crosslinking of sulfur radicals promoted the yield of residues. J. Zhang [66] et al. synthesized a new type of hyperbranched polymer (ITA-HBP) by employing tartaric anhydride and DOPO. ITA-HBP was used to prepare flame-retardant EP. Experimental results showed that the introduction of ITA-HBP significantly improved the toughness and flame retardancy of EP. When the phosphorus content is 0.26 wt.%, the LOI of ITA-HBP/EP increased to 36.3%, and the UL-94 rating reached V-0 level. The pk-HRR of the EP composites reduced. The tensile strength, flexural strength, impact strength, and fracture toughness of ITA-HBP/EP enhanced with the increasing content of ITA-HBP. It is worth mentioning that ITA-HB could work in both gas and condensed phases. Recently, a new phosphorus-containing polyphosphate flame retardant (PFR) was synthesized by Carja [67] et al. and they prepared a series of PFR-EP semi-interpenetrating polymer network (SIPN) composites (Figure 2). Experimental results showed that the flame retardancy improved as the PFR concentration increased. While the content of PER reached 22.28 wt.%, the flame-retardant EP exhibited a high LOI value of 42% and passed UL-94 V-0 rating. The char yield was up to 40.33% and the ignition time (TTI), pk-HRR and THR decreased simultaneously. The phosphorus-oxygen bond on molecular chain of flame retardant PFR could promote compatibility and adhesion with the matrix resin, which extremely enhanced the flame retardancy of EP.

The flame performance of thermosetting polymer based on salts of dialkylphosphinate was widely reported [68,69]. X. Liu [70] et al. studied flame retardant EPs that prepared by using phosphinate as a flame retardant. Aluminum diethylphosphinate [Al (DEP)] and aluminum methylphosphinate [Al (MEP)] (Figure 3) were added into EP, respectively. The results suggested that Al (MEP) and Al (DEP) exhibited high flame-retardant effect on EP. The modified EP could pass UL-94 V-0 rating and the LOI value reached 32.2% [Al (MEP)] and 29.8% [Al (DEP)] at the 15 wt.% content of aluminum phosphinate. Both Al (MEP) and Al (DEP) could promote the char formation and enhance the flexural modulus of the EP, while the flexural strength was sacrificed. Compared with Al (DEP)/EP, Al (MEP)/EP exhibited better flame-retardant performance but lower flexural properties.

In addition, phosphine oxide flame retardants have gained wide attention from scientists and engineers. J. Sun [71] et al. synthesized bis-phenoxy (3-hydroxy) phenyl phosphine oxide (BPHPPO) and EP composite (BPHPPO-EP) was prepared with BPHPPO as raw material (Figure 4). The results showed that the introduction of BHPPO imparts flame retardancy to EP. The LOI value was 34%, and the char yield was up to 51.8%. Moreover, the phosphorus-containing EP produced little fumes when burned. However, the low tensile strength of BPHPPO-EP may limit its application in certain areas. V. Cadiz [72] et al. prepared a novel EP based on diglycidyl ether of (2,5-Dihydroxyphenyl) diphenyl phosphine oxide and the effect on the thermal stability and flame retardancy of the material was investigated. They observed considerable improvements in flame-retardant performance that adding phosphorus. The char yield in air improved with the increasing of phosphorus content, but there was no significant difference between phosphorus-containing resin and non-phosphorus resin in the char yield under nitrogen.

#### 2.1.3. DOPO

DOPO-based flame retardants are important intermediates in organophosphorus flame retardants. DOPO and its derivatives have been increasingly attractive for researchers due to its high thermal stability, hydrolysis resistance, oxidation resistance and high flame-retardant efficiency [73,74,75]. According to previous reports, DOPO-based flame retardants can significantly enhance flame-retardant performance but impair the mechanical properties of EPs [76,77]. It releases free radicals to interrupt free radical combustion in the gas phase, and dehydration to carbon in the condensed phase [78]. Numerous works have been made in the molecular design and synthesis of DOPO derivatives [79,80].

DOPO could achieve better flame-retardant effect in a small amount of addition. It is frequently observed that adding DOPO-compounds could decrease the glass transition temperature (T_g_) and sacrifice the mechanical properties. Perret [81] et al. synthesized two phosphorus-containing flame retardants (DOPI, DOPP) (Figure 5) by reaction of tris (2-propylene ester oxyethyl) isocyanurate and pentaerythritol tetra acrylate with flame retardant DOPO, respectively, and applied them to EP and CF reinforced materials. DOPP and DOPI worked by suppressing the flame in the gas phase. In the condensed phase, a part of phosphorus was bound to the char network by decomposition reactions, which increased the char formation. When the phosphorus content of the flame-retardant system was 0.6 wt.%, the LOI reached 45.3% (DOPI) and 47.7% (DOPP), respectively, and both achieved the important criterion UL-94 V-0. 

A new DOPO derivative named ABD based on DOPO and acrolein was synthesized (Figure 6) by L. Qian [51] et al. and applied it to EP. The experimental results showed that the EP with 3 wt.% ABD exhibited the best flame retardancy: LOI obtained 36.5% and passed UL-94 V-0 rating. The introduction of ABD reduced the pk-HRR, THR and TSP of modified EP. Compared with the flame retardant effect of DOPO, the flame retardancy of ABD can be ascribed to the combination of the quenching action of the phosphorus phenanthrene group in the gas phase of ABD and the carbonization of hydroxyl groups. 

T. Wang [82] et al. synthesized epoxy composites with DOPO and 1,10-(methylenedi-4,1-phenylene) bismaleimide (BDM). The test results indicated that BDM and DOPO have a synergistic effect on the flame retardancy of EPs. With the synergistic effect of BDM and DOPO, the av-HRR, pk-HRR and THR was reduced as the load of BDM increased but simultaneously led to an increase in LOI. Moreover, epoxy composites reached the UL-94 V-0 grade, and the carbon residue increased considerably at high temperature. 

A DOPO-based tetrazole derivative (ATZ) was synthesized (Figure 7) and integrated into the EP matrix as co-curing agent [83]. The addition of ATZ significantly improved the flame retardancy of EPs. The LOI value of 33.7% and UL-94 V-0 rating were obtained when the content of ATZ was as low as 6 wt.%. Significant reduction in PHRR and THR compared to neat EP. Furthermore, ATZ could act in both gas and condensed phases. In the gas phase, flame retardance was mainly achieved by the quenching effect of phosphorus radicals and the suppression of flame by inert gas. The flame retardance of the condensed phase was due to the blocking effect of the carbon layer formed by the phosphoric acid derivative and the epoxy group.

A DOPO-based pyrazine derivative (DHBAP) was synthesized via a two-step addition reaction of 2-aminopyrazine, 2-hydroxybenzaldehyde and DOPO, its fire performance tested on EPs [84]. A V-0 rating in the UL-94 test and an LOI value of 34% could be achieved at a loading of 8 wt.% of DHBAP. In addition, lower HRR, THR, smoke produce rate (SPR) and TSP was observed for EP/8% DHBAP compared with neat EP. DHBAP was active in two-phase, which is similar to the flame-retardant mechanism of ATZ.

#### 2.1.4. Phosphorus/Silicon Synergistic Flame Retardants

In addition to the above phosphorus-containing flame retardants, phosphorus/silicon compounds also have aroused much attention [85,86,87]. Silicone flame retardant is a non-halogen flame retardant with high efficiency, anti-melting and smoke suppression, which can effectively improve the processing performance, mechanical properties and heat resistance of EP. Phosphorus-containing flame retardants have a significant synergistic flame retardant effect while combined with silicon compounds, and silicon can improve the thermal stability of carbon layer catalyzed by phosphorus [88,89]. In recent years, the phosphorus-silicon synergistic flame-retardant system have been intensively investigated by many researchers [90,91].

Currently, phosphorus-silicon synergistic modification has become an increasingly popular option for improving flame retardancy of EPs. R. Yang [92] et al. prepared a series of flame retarded EPs by using a novel polyhedral oligomeric silsesquioxane (POSS) combined with DOPO as flame retardant. The results indicated that the introduction of DOPO-POSS could significantly improve flame retardancy and thermal stability of EP. In addition, different contents of DOPO-POSS in EP exhibited different flame-retardant properties. EPs with a 2.5 wt.% DOPO-POSS adding have the best flame retardancy: LOI of 30.2% and a UL-94 rating of V-1 (t_1_ = 8 s and t_2_ = 3 s) were obtained. The flame-retardant performance weakened when the DOPO-POSS content increased from 3.5 wt.% to 10 wt.%. Q. Kong [93] et al. synthesized layered phenyl zirconium phosphate (ZrPP) by mixed solvothermal technology for the first time and incorporated it with POSS to flame-retard EP. It was found that introduction of POSS reduced pk-HRR and the addition of ZrPP prolonged the TTI. Moreover, ZrPP and POSS exhibited a synergistic effect in improvement of flame retardancy and thermal stability. The insertion of POSS molecules into ZrPP improved the dispersion of the system and nano-reinforcement of the composite. 

It is rarely reported that triphosphates and epoxy-functional polysiloxanes could synergistically modify EPs. S. Li [87] et al. synthesized phosphorous triamide (PTA) and epoxy-functional polysiloxane (EFP) (Figure 8) to modify the EP in order to provide the EP with flame retardancy and mechanical properties. The results suggested that the char yield of modified EP was 16.6% at 700 °C, and its LOI value reached 30.2%, which was 50% higher than neat EP. The UL-94 rating reached V-1. Furthermore, the addition of PTA and EFP notably improved the failure strain of the EP and maintained other original properties. Analysis of the test results of mechanical properties showed that that the increased free volume in the curd network and outstanding flexibility of the –Si–O–Si skeleton led to this phenomenon.

In addition to the above, there is a new type of flame retardant with P-O-Si bond in the main chain. Y. Huang [94] et al. synthesized a novel silicon/phosphorus-containing hybrid (SPDS) by polycondensation of diphenylsilanediol and spiropentaerythritol di (monophosphate). Furthermore, SDPS was collaborate used with mono (4, 6-diamino-1,3,5-triazin-2-aminium) mono (2,4,8,10-tetraoxa-3,9-di-phosphaspiro [5.5] undecane-3, 9-bis (olate) 3, 9-dioxide) (SPDM) (Figure 9) to flame retard EP. The study found that SDPS and SPDM could play a synergistic flame-retardant effect in the modified EP system. The synergistic of SDPS and SPDM provided high LOI value and high char yield for the EP. However, the LOI value of the composites was related to the ratio of SPDS and SPDM. Moreover, the introduction of SPDS and SPDM in EP significantly reduced the PHRR of composites. Test results of TGA, TG-FTIR and SEM showed that the dense honeycombed carbonaceous structure formed by the thermal degradation of SDPS/SPDM and inhibition of flammable gas release improved the flame retardancy of EP.

#### 2.1.5. Phosphorus/Nitrogen Synergistic Flame Retardant

Currently, phosphorus/nitrogen synergistic flame retardants have become a new focus due to their unusual flame retardancy and thermal stability [95]. Nitrogen-containing flame retardants generate inert gases during combustion to suppress flame combustion. However, the flame retardant efficiency became less effective when used alone due to the limitation of nitrogen content [96]. Generally, nitrogen and phosphorus are used in combination to improve flame retardant efficiency. Cyclophosphazene derivatives have been widely studied for their flame retardancy and reactivity in the field of flame retardants. Phosphazene compounds obtained by nucleophilic substitution reaction with hexachlorocyclotriphosphazene are a class of compounds containing P = N in the molecule structure [97,98]. Due to the synergistic flame retardance of phosphorus and nitrogen, the addition of cyclotriphosphazene to EP could significantly improve the thermal and flame retardancy properties of EP, further improving the safety of EP in various application fields.

Cyclotriphosphazene derivatives have lower flame retardance due to lower phosphorus content. Therefore, it is necessary to add other additives to enhance its flame retardancy. P. Liu [99] et al. synthesized a cyclophosphazene compound (CP-6B) (Figure 10) containing phosphorus, nitrogen, and boron, and its fire performance tested on EPs. The results showed that CP-6B could form a dense carbon layer on the surface and generate non-combustible gas during combustion. An LOI value of 32.3% and a UL-94 V-0 rating was obtained in the case of with an amount of 7 wt.% of CP-6B; the pk-HRR and THR were decreased significantly.

A series of flame-retardant epoxy thermosets with cyclolinear cyclotriphosphazene-linked (CL-CPTN) structure were synthesized via a three-step synthetic route (Figure 11), and their flame retardancy was investigated [100]. The results showed that the epoxy thermosets possessed thermal stability and flame retardancy. All epoxy thermosets passed the UL-94 V-0 test, and the LOI value could reach more than 30%. The thermooxidative reaction of phosphazene compounds promoted the formation of protective carbon layer, further improving the flame retardancy of epoxy thermosets. It was worth mentioning that cyclotriphosphazene-linked epoxy thermosets exhibited better mechanical properties than neat EP. Cyclolinear structured flame-retardant EP based on phenol-substituted cyclotriphosphazene [101] and spirocyclic phosphazene-based EP (SP-epoxy) [102] were synthesized in their latter work also obtained similar results. Moreover, they prepared SP-epoxy/graphene nanocomposites by the exfoliation of graphite platelets and thermal curing process. The experimental results exhibited that the introduction of graphene had a positive effect on the tensile properties, flexural properties and electrical conductivity of composites.

A novel flame retardant bisphenol s-bridged penta (anilino) cyclotriphosphazene (BPS-BPP) was synthesized (Figure 12) and applied on EP at different loadings [103]. The LOI value of 29.7% and UL-94 V-1 rating was achieved when the content of BPS-BPP was 9 wt.%, pk-HRR and TSP was decreased significantly. BPS-BPP acted via the two-phase mechanism. BPS-BPP released aniline and formed cross-linked cyclotriphosphazene structure below 450 °C, leading to a significant increase in char residue. Above 450 °C, NH_3_ and phosphorus components produced by the decomposition of cyclotriphosphazene were released into the gas phase, which suppressed the propagation of the flame. 

A flame retardant (CTP-DOPO) with cyclophosphazene structure and DOPO was synthesized and its fire performance tested on EPs [104]. When the content of CTP-DOPO was 10.6 wt.%, UL-94 V-0 rating and a high LOI value of 36.6% were reached, and pk-HRR, THR and TSP decreased significantly. CTP-DOPO could promote the formation of carbon layer, further improving the flame retardancy and thermal stability of epoxy resin.

J. Mu [105] et al. synthesized a novel phosphazene-based flame retardant (HPCTP) and incorporated with octapropylglycidylether POSS (OGPOSS) (synthesized by Liu [106] et al.) into EP. The results showed that EP composites cannot pass the UL-94 rating when the OGPOSS content was 15 wt.%. However, the addition of HPCTP could make the EP composites reach UL-94 V-0 grade. When the HPCTP content was 15 wt.%, the pk-HRR and THR of the composites were reduced by 61% and 48% compared to neat EP, respectively. In addition, HPCTP formed closed honeycomb pore structure in the EP composite during combustion, which could inhibit heat transfer and flame diffusion, further enhancing flame retardancy of the EP composites.

The flame retardant performance of the phosphorus-containing flame retardants mentioned in the article is shown in Table 2.

### 2.2. Carbon-Based Materials

Carbon-based materials flame retardants have become a popular issue due to their possessing many advantages, such as being wide source, non-toxic, environmentally friendly, etc. [107]. They have broad application prospects as halogen-free flame retardants in polymers [108]. Carbon-based materials that act as flame retardants include graphene, carbon nanotubes (CNTs), expandable graphite, etc. They could improve the flame-retardant performance of polymers by increasing the thermal stability of the polymer, extending TTI, and reducing heat release [109]. However, the flame retardancy of polymer could not be improved markedly when carbon materials were used as an independent flame retardant. Therefore, they need to be compounded with other materials to achieve high performance [110].

#### 2.2.1. Graphene

Graphene, which has a stable two-dimensional honeycomb monolayer lamellar crystal, is formed by the hybridization of carbon atoms on graphite flakes [111]. It has a better flame-retardant effect because of its larger specific surface area and layered barrier effect. Graphene shows good electrical conductivity, thermal stability and mechanical properties. Graphene is obviously a promising flame retardant additive, but flame-retardant efficiency is unsatisfactory when neat graphene used alone. In order to expand the application range of graphene, functional groups are often used to modify its surface. A lot of researches on graphene have been conducted by many researchers [112,113].

As an important functional metal oxide, manganese oxide is used in flame retardation due to its unique properties. It was also found that cerium-containing compounds also have a flame suppression effect. Recently, it has been reported that these two components can work synergistically and may provide unexpected flame retardancy to polymers. S. Jiang [114] et al. prepared a Ce-doped MnO_2_-graphene composite sheet by electrostatic interaction between Ce-doped MnO_2_ and graphene sheets (Ce-MnO_2_-GNS). This work focused on the effect of the introduction of Ce-doped MnO_2_-graphene hybrids on the fire hazard behavior of epoxy nanocomposites. The results exhibited that GNS and Ce-MnO2 played a synergistic role in improving the thermal stability and flame retardancy of Ce-MnO_2_-GNS-EP. Compared with neat EP, the mass decomposition temperature (T_50%_) and the maximum mass decomposition rate (T_max_) of Ce-MnO_2_-GNS-EP were increased by 4.6, 8.8 and 8.9, respectively. The carbon residue rate increased by 467%, pk-HRR, THR, TSR decreased by 53.7%, 35.5% and 41.2%, respectively, and the release of organic volatiles and toxic gases decreased. 

GNSs have drawn wide attention from the field of fire safety due to their unique structure, outstanding thermal and mechanical properties. However, the oxidation reaction of GNS at high temperature weakens the barrier effect. Therefore, it is necessary to modify GNS to prevent oxidation. GNS surface modification with layered MoS_2_ seems to be a good choice. D. Wang [115] et al. studied the flame retardant effect of nano-flake graphene GNS modified by MoS_2_ on EP. Compared with neat EP, the thermal stability of MoS_2_/GNS/EP composites with 2.0 wt.% MoS_2_/GNS mass fraction was significantly improved, initial decomposition temperature increased by 53 °C (in air); pk-HRR and THR decreased by 45.8% and 25.3% respectively, and the TSR was reduced by 30.5%. In recent years, layered double hydroxide (LDH) has aroused great concern. Both LDH and GNS have significant barrier effects. The combination of the LDH and GNS could promote the flame-retardant efficiency of graphene. X. Wang [116] et al. prepared nickel-iron double hydroxide Ni-Fe LDH/graphene GNS by self-assembly. The fire safety performance of the composites (Ni-Fe LDH/GNS/EP) which were prepared by introducing Ni-Fe LDH/GNS was studied. When the mass fraction of Ni-Fe LDH/GNS was 2.0 wt.%, the initial decomposition temperature increased by 25 °C, the TTI delayed by 21 s, pk-HRR, THR, average mass loss rate (AMLR) and fire growth rate index (FIGRA) decreased significantly by 60.8%, 60.9%, 53.5%, 63.9%, respectively. The synergistic flame-retardant mechanism of Ni-Fe LDH and GNS included both gas phase and condensed phase.

#### 2.2.2. Carbon Nanotubes

CNTs are considered as candidate flame retardant additives because the mechanical, electronic and flame-retardant properties [117]. The application of CNTs in flame retardants has attracted great interest among researchers [118]. During the combustion of the polymer, the CNTs form a barrier carbon layer with a network structure on the surface. The barrier layer has the function of isolating heat, oxygen and preventing the combustion of combustibles from escaping into the air [119]. The introduction of CNTs in polymers could impart flame retardancy and mechanical properties, but CNTs are difficult to disperse uniformly when blended with polymers, which leads to the degradation of polymer mechanical properties.

Buckypapers (carbon nanotube membranes) are free-standing mats of entangled carbon nanotube ropes, which can be produced by the filtration of CNT suspensions. The flame-retardant function of buckypaper is to form a protective layer composed of CNTs or nanofiber sheets, which limits the contact of the decomposed gas and oxygen. C. Zhang [120] et al. applied single-walled carbon nanotube (SWCNT) and multi-walled carbon nanotube (MWCNT) membranes (buckypaper) and carbon nanofiber (CNF) papers to the surface of epoxy carbon fiber composites. Researchers found that SWCNT buckypapers and CNF papers did not exhibit extraordinary improvement on flame retardancy. In contrast, MWCNT buckypapers could significantly reduce pk-HRR by more than 50% and reduce combustion fumes by half. They attributed the flame retardancy of the composite to its high thermal-oxidative stability and dense network. The MWCNT-based buckypaper served as an efficacious fire protection board to reduce the heat, smoke, and toxic gases produced in the process of fire combustion. 

CNTs have been widely used in nano-reinforced various polymer matrices due to their many characteristics, which has attracted increasing interest for researchers in the field of polymer and materials. C.-L. Chiang [121] et al. introduced the CNTs which were functionalized with vinyl triethoxysilane (VTES) (named as VTES-CNT) into EPs and prepared EP composites by sol-gel reaction. Test results suggested that EP composites showed the best flame retardancy at a VTES-CNTs content of 9 wt.%. Both T_g_ and char yield of composites was increased by more than 45%. The LOI value reached 27% and passed UL-94 V-0 rating. 

The dispersion of CNTs in polymers is a key factor affecting the flame retardancy of CNTs. J. Liu [122] et al. grafted molybdenum-phenolic resin (Mo-PR) onto the surface of MWCNTs to obtain modified MWCNTs (CNT-PR) (Figure 13), and introduced it into EP. The results showed that the introduction of CNT-PR improved the dispersion of CNTs, thereby enhancing the flame retardancy and mechanical properties of nanocomposites. The composite containing 3 wt.% CNT-PR and 8 wt.% melamine had a LOI of 29.5% and passed the UL-94 V-0 test.

CF-reinforced epoxy composites have been widely used in recent years due to their properties. G.Q. Zhu [123] et al. synthesized a series of CNTs/CF/EP composites and investigated the effect of CNTs on the flame retardancy and mechanical properties of CF/EP composites. The results showed that the addition of CNTs could improve the flame retardancy and smoke suppression of CF/EP composites. Overall, 0.7 wt.% CF/0.7 wt.% CNTs/EP composite had the smallest pk-HRR, which was 34.7% and 12.1% lower than neat EP and 0.7 wt.% CF/EP composite, respectively. The TSP of 0.5 wt.% CF/0.5 wt.% CNTs/EP composite was 43.0% and 37.7% lower than neat EP and 0.5 wt.% CF/EP, respectively. In addition, analysis of mechanical test results revealed that CNTs and CF exhibited synergistic effects in improving EP mechanical properties. 

The melamine salts of pentaerythritol phosphate are promising flame retardants due to their good compatibility with polymers and high flame retardant efficiency. A functionalized multi-walled CNTs was synthesized by grafting pentaerythritol phosphate melamine salt (PPMS) onto the surface of CNTs (named as PPMS-CNTs) and its fire performance was tested in EPs [124]. The results showed that the LOI and UL-94 rating of PPMS-CNT/EP composites improved compared to neat EP. In the condensation phase, PPMS-CNT could promote the formation of the carbon layer, and improved the barrier effect and oxidation resistance of the carbon layer, thereby improving the flame retardancy and smoke suppression of the EP composites. In the gas phase, PPMS-CNT decomposed at a high temperature to generate P–O· radicals, H_2_O and NH_3_, which suppressed flame combustion and further achieved flame retardant and smoke suppression effects.

#### 2.2.3. Expandable Graphite

As one of the new generations of intumescent flame-retardant additives, expandable graphite (EG) possesses many advantages, such as wide source, non-toxic, environmentally friendly, etc. [125]. EG could expand on the surface of the polymer matrix to form an insulating layer at high temperatures, providing flame retardancy to the polymer [126]. However, since the carbon element has strong stability and poor surface activity, it is difficult to be uniformly disperse the expandable graphite in the resin which lead to the degradation of the performance of the resin. Therefore, EG needs to be modified with other materials to achieve flame retardancy and maintain the mechanical strength [127]. The flame-retardant effect of EG and its derivatives in EPs have been widely concerned by many authors [128,129].

EG has been touted as an impactful intumescent flame-retardant additive, but the mechanical properties of the composite will be decreased by excessive addition. Therefore, the proportion of EG in the compound becomes important. A. Mamani [130] et al. evaluated the effects of EG loading on the mechanical properties, flame retardancy and thermal properties of epoxy-aliphatic amine system. It was found that the flame-retardant performance of the EP composites improved but the mechanical properties decreased with the increase in EG content. Therefore, the optimum value of the EG concentration in the epoxy/EG composite was determined in consideration of both thermal and mechanical properties. The results showed that the composite with 2 phr EG exhibited the maximum balance between heat, flame retardancy and mechanical properties. Jatropha curcas oil-based alkyd resin has the advantages of multifunctional structure and performance. P. Gogoi [131] et al. used jatropha curcas oil and EG as raw materials to prepare jatropha curcas oil/EP/EG bio composites. The best thermal stability of the jatropha curcas oil/EP/EG composites appeared when 5 wt.% EG was added. Experimental analysis showed that EG was uniformly dispersed in the polymer, which indicated that EG was compatible with the jatropha curcas oil-based alkyd and epoxy resin blend. The LOI value as high as 41%, which was 2.3 times higher than that of jatropha curcas oil/EP composites.

The flame-retardant performance of the carbon-based materials mentioned in the article is shown in Table 3.

### 2.3. Silicon Flame Retardants

Silicon-based flame retardants have been a research hotspot in the field of halogen-free flame retardant in recent years [132]. Silicon-based flame retardants mainly include siloxanes, polysiloxanes, silicas, etc. [133]. The well-known advantages of silicon-based flame retardants, such as halogen-free, low-smoke, low-toxicity characteristics have attracted wide attention from researchers [134,135]. The main flame retardant mechanism of silicon-based flame retardant is to form a cracked carbon layer and improve the oxidation resistance of the carbon layer [136]. Adding silicon-based flame retardants to polymers not only enhance the flame retardancy of the polymer, but also improve its mechanical properties and heat resistance [137]. However, some silicon flame retardants need to be used in conjunction with other flame retardants or compounds to achieve the desired flame retardant effect due to their lower flame retardant efficiency.

#### 2.3.1. Siloxane

Siloxanes are important silicon flame retardants with Si-O-Si bond as the main chain and alkyl, alkyl substituent and phenyl as the side chain. In recent years, siloxanes have attracted widespread attention as efficient, non-toxic, low smoke emission and environment-friendly flame retardants [138,139]. Siloxanes with low surface could migrate from the polymer to the surface during combustion, and react with the polymer to form a dense and stable oxygen-insulating carbon layer, which not only prevents the combustion decomposition products from escaping, but also inhibits the thermal decomposition of the composites [140,141].

A novel siloxane-containing cycloaliphatic epoxy composite 1,3-bis[3-(4,5-epoxy-1,2,3,6-tetrahydrophthalimido) propyl] tetramethyldisiloxane (BISE) was synthesized (Figure 14) [142]. BISE had mechanical properties and thermal stability compared to conventional cycloaliphatic EPs. Moreover, BISE had amazing dielectric properties and moisture resistance due to the low polarity and hydrophobic nature of the siloxane segments in the epoxy backbone. The imide group possess good thermal stability and mechanical strength, and it has good compatibility with EP at high temperature, which has drawn a lot of attention from scientists and engineers. A novel siloxane- and imide-modified EP which cured with siloxane-containing dianhydride was synthesized [143]. The siloxane group and the imide group greatly improved the toughness, thermal stability and mechanical properties of the EP. It was worth noting that the T_g_ values of these modified systems were higher than 160 ° C. The reason for the high thermal stability was that the siloxane group was converted into flake silica as char residual during pyrolysis, which effectively inhibited heat transfer in the polymer.

In order to achieve higher flame retardant effect, siloxanes are often used in combination with other highly efficient flame retardant additives. A series of novel macromolecules containing phosphinophenanthrene/phenylsiloxane bifunctional groups (DDSi-n) were synthesized (Figure 15) and their fire performance tested on EPs [144]. The results suggested that the introduction of DDSi-n macromolecules enhanced the flame retardancy and mechanical properties of EPs due to the synergistic flame retardant effect of phosphinophenanthrene and phenylsiloxane groups in EPs. The LOI value of 35.9% and UL-94 V-1 rating was obtained when the content of DDSi-1 was 8 wt.%. The LOI value of the epoxy composite decreased with the increase in the degree of polymerization of DDSi-n at the same amount of addition, while the UL-94 level enhanced. Both pk-HRR and THR decreased significantly with the increase in DDSi-1 content. The impact strength of DDSi-n/EP increased as the content of DDSi-n increased or the degree of polymerization decreased. In their later work, two cluster-like molecules were synthesized using DDSi-1 (TriDSi and TetraDSi), and added it to EPs [145]. The incorporation of TriDSi and TetraDSi simultaneously enhanced the flame retardancy and mechanical properties of EPs. When the additive amount was 6 wt.%, the epoxy composites passed the UL-94 V-0 test, the LOI values could reach 35.2% (TriDSi) and 36.0% (TetraDSi), and the impact strength increased by 133% (TriDSi) and 123% (TetraDSi), respectively, and the pk-HRR and THR reduced compared with neat EP. The analysis of comprehensive experimental results showed that the synergistic effect of phosphophenanthrene and siloxane groups and the segmer-aggregation effect of cluster-like molecules provided flame retardancy and mechanical properties of EP.

#### 2.3.2. Silica

Silica is one of the commonly used fillers for modifying EP. The introduction of silica in polymers could promote the formation of carbon layers and improve the oxidation resistance, which not only provides the polymer with flame retardancy, but also improves the polymer’s processability, mechanical properties and heat resistance [146,147]. The preparation of novel silicon-based materials is one of the important methods to develop environment-friendly flame-retardant materials.

Spherical silica has aroused great interest due to the advantages of high loading, high fluidity and low thermal expansion coefficient. S.-E. Hou [148] et al. prepared spherical silica powders by using homogenizer and added it to EP as an additive. The effects of different percentages of spherical silica on thermal stability, coefficient of thermal expansion and mechanical properties were investigated. The initial decomposition temperature and mechanical properties of the composites were significantly improved by adding silica. Moreover, the composites exhibited maximum thermal stability and mechanical properties at a spherical silica content of 30 wt.%.

A green multi-element silica derivative (HM-SiO_2_@CS@PCL) was synthesized by coating chitosan (CS)/phosphorylated cellulose (PCL) onto the surface of hollow mesoporous silica (HM-SiO_2_) (Figure 16), and integrated into the EP matrix as flame retardant [149]. The results showed that HM-SiO2@CS@PCL exhibited high flame retardant efficiency. The char residue at 700 °C increased from 5.0% to 17.8%. The pk-HRR and TSP decreased by 51% and 18.7%, respectively. The flame retardancy of EP/HM-SiO2@CS@PCL was improved due to the formation of a dense carbon layer, which acted as a protective barrier.

A. Afzal [150] et al. synthesized a series of epoxy-silicon nanocomposites containing 0–20 wt.% silica via two-step sol-gel process, and epoxy-silane coupling agent (3-glycidoxypropyl) trimethoxysilane (GLYMO) was added to EP nanocomposites. It was found that the thermal degradation rate of epoxy nanocomposites slowed down significantly with increasing silica content, the char residue and T_g_ was increased. The EP nanocomposites possessed thermal stability due to the addition of silica, and the presence of GLYMO enhanced the organic compatibility of epoxy nanocomposites, further improving the thermal stability.

Three different phosphorus-containing silicas were prepared from di-sodium hydrogen orthophosphate, orthophosphoric acid, hypophosphorous acid and sodium silicate respectively, and added these three compounds to the epoxy composite [151]. The results showed that the addition of phosphorus-containing silica could increase the carbon residue of 10–14 wt.% at 650 °C. The LOI value rises marginally due to lower phosphorus content. However, the flame spread slowly due to the formation of coal char with foam structure. Cuong M.V. [152] et al. prepared phosphorus-containing epoxy soybean oil (DOPO-J-ESO) and rice husk-based silica (RH-SiO_2_), used in combination with EPs and subsequently investigated for their flame retardancy and mechanical properties. The results exhibited that the introduction of RH-SiO_2_ enhanced the flame retardancy and mechanical properties of epoxy composites. With the addition of 20 wt.% RH-SiO_2_/10 wt.% DOPO-J-ESO_,_ a V-0 rating in the UL-94 test and an LOI value of 35.9% could be achieved, the PHRR was reduced by 46.1% compared to neat EP, and the impact strength was increased by 90.1%. Analysis of the results of the energy-dispersive X-ray spectroscopy (EDS) revealed that the improvement in flame retardancy was due to the carbon layer formed during the combustion process, and the polymer oxygen (PO) and polymer peroxy (POO) radicals formed by the decomposition of DOPO-J-ESO react directly with H and OH to form completed molecules.

#### 2.3.3. POSS

POSS are nano compounds with a three-dimensional size of 1 to 3 nm, where the distance between Si atoms is 0.5 nm and the distance between R groups is 1.5 nm. Their general formula is Si_n_O_3n/2_R_n_ (usually referred to as T_n_R_n_) [153]. Among them, the most common form (Si_8_O_12_R_8_ or T_8_R_8_) is a cage frame composed of 8 silicon corner atoms and 12 oxygen atoms. Each silicon corner atom may be connected to a group (R), and the properties and number of R groups can be selectively controlled according to different requirements for material properties [154]. There has been a growth interest in application of POSS due to their potential for modification [155,156]. The structure of POSS combines many advantages of silica and siloxane, including thermal stability, chemical stability, mechanical properties, low toxicity, solubility, ease of modification and flame retardancy, etc. [153]. POSS-based polymers could form a carbon layer on the surface at high temperatures to prevent oxygen and heat transfer [157]. Currently, various EP/POSS systems have been reported [158,159].

A series of phosphorus containing POSS was synthesized via the reaction of octa vinyl POSS with diphenylphosphine (DPP), diphenylphosphine oxide (DPOP), and DOPO respectively (Figure 17) [160]. Flame retardant epoxy composites with a flame retardant content of 5 wt.% were prepared using DOPO-POSS, DPOP-POSS and DPP-POSS, respectively. The results showed that all three flame retardants had the ability to improve the flame retardancy of EPs. The LOI value reached 33.2% (EP/DPP-POSS), 29.3% (EP/DPOP-POSS) and 30.0% (EP/DOPO-POSS), respectively, EP/DPP-POSS passed UL-94 V-0 test, EP/DPOP-POSS and EP/DOPO-POSS passed UL-94 V-1 test. The pk-HRR and THR significantly reduced and the residues increased from 3.5% (neat EP) to 20.2% (EP/DPP-POSS), 17.9% (EP/DPP-POSS), 19.1% (EP/DPP-POSS), respectively. These flame retardants were proposed to be active in both the condensed and gas phases. The flame retardant effect was achieved by the release of phosphorus volatiles in the gas phase and promoting charring in the condensed phase.

A novel halogen-free flame retardant (ODMAS) containing DOPO and POSS was synthesized (Figure 18), and used in combination with EPs and subsequently investigated for their flame retardancy [161]. Experimental results showed that EP/ODMAS composites exhibited fire resistance. The LOI value of 35.5% and UL-94 V-0 rating was obtained even when the content of ODMAS was as low as 5 wt.%. The char yield of EP/DOMAS composites increased with the increase in the content of ODMAS. Furthermore, the addition of ODMAS to EP matrix improved the mechanical properties due to the good solubility of ODMAS in EP.

A flame-retardant additive (POSS-bisDOPO) containing POSS, DOPO and polyoxymethylene (POM) was synthesized via Kabachnik–Fields reaction and its fire performance tested on EPs [162]. POSS-bisDOPO had a special surfactant structure which gave it self-assembly ability. The introduction of POSS-bisDOPO in EP improved the flame retardancy and mechanical properties of epoxy resins due to its outstanding self-assembly ability. The LOI value of the composites was 34.5% when the content of flame-retardant was 20 wt.%. In addition, the composites produced a dense char with a porous structure on the surface during combustion, further improving the flame retardancy and thermal stability of the composites.

In addition, POSS can be grafted and copolymerized with other materials or monomers to improve the thermal stability of the composites at the molecular level [163]. Y. Zheng [164] et al. synthesized liquid-like trisilanol isobutyl polyhedral oligomeric silsesquioxanes derivative (L-POSS-D) and prepared L-POSS-D/epoxy nanocomposites. The T_g_ of nanocomposites with 2.0 wt.% L-POSS-D was 47.7 °C higher than neat EP. L-POSS-D also improved the impact toughness and flexural strength, which were 157.7% and 22.7% higher than neat EP, respectively. The improvement of the mechanical properties was due to the great interaction between L-POSS-D and EPs and the reduction in the nano-fillers’ volume fraction in composites. G. Liang [165] et al. synthesized a series of inorganic-organic diglycidyl ethers of bisphenol A/octaamino (aminopropyl) silsesquioxane composites. The results showed that the addition of POSS could obviously enhance the thermal stability of the EP. The T_g_ of the composites increased with increasing content of POSS until its content exceeded 10 wt.%. The char residue of EP composite monotonously increased with increasing content of POSS. The increase in carbon residue content was due to the Si-O-Si structure of POSS leading to higher inorganic components and higher residual carbon.

The flame retardant performance of the silicon flame retardants mentioned in the article is shown in Table 4.

### 2.4. Nanocomposites

In recent years, nanocomposites have been extensively researched and developed [166,167]. Due to the surface effects, volume effects, and quantum size effects of nanoparticles, polymer/nanocomposites overcome many shortcomings of conventional polymer composites, and meets the needs of developing functional new materials with high performance [168,169]. The incorporation of nanocomposites not only promote the strength and toughness, but also provide heat resistance, barrier properties and flame retardancy of polymer. Among them, montmorillonite (MMT) is a nano-flame retardant material with many properties [170]. Due to the unique structure of the MMT nanocomposites, the polymer can be intercalated between the silicate sheets by intercalation and recombination to achieve the nano-scale combination of the polymer and the layered silicate. The MMT nanocomposite forms a dense barrier carbon layer on the surface which could reduce the heat of combustion to the combustion portion and the diffusion of decomposition products into the flame region, it also inhibits the generation of volatiles by insulating heat to achieve a flame-retardant effect [171].

It has been reported that the addition of organically modified montmorillonite (OMMT) to EP could improve the performance of thermal and flame-retardant. W. Yan [172] et al. mixed phenethyl bridged DOPO derivative (DiDOPO) with OMMT and added it into EPs. Experiment result showed that the composite exhibited flame retardancy when the addition of 3.5 wt.% DiDOPO and 3.5 wt.% OMMT. The composite passed UL-94 V-0 rating and LOI was 32.2%. The flame-retardant performance of the EP/DiDOPO/OMMT composite was promoted by the barrier effect of the OMMT and the flame-retardant effect of DiDOPO in the gas phase.

In addition to MMT, CNTs are also one of the most widely used fillers. The effect of adding fluorinated MMT/MWCNT additives on EP was investigated by Y.-S. Lee [173] et al. MMT and MWCNT were fluorinated to enhance their dispersion in EPs. The thermal insulation of MMT and the high specific heat of MWCNT hindered the heat transfer of EP. The char yield increased from 9.1% to 15.4% and LOI grew from 21% to 31% due to the influence of flame retardants. Moreover, the thermal stability was significantly improved by fluorination of MMT and MWCNT. MMT particles usually present in polymer materials in micron sizes; it is difficult to obtain true nanoscale dispersions of clay due to its immiscibility with the polymer. W. Zhang [174] et al. used two different methods to add APP and MMT into EPs (APP-MMT: in situ addition; APP + MMT: physical mixing). It was found that both of these methods made EP obtain a higher value of LOI, which were 28% and 30%, respectively. Both of them passed the UL-94 V-0 test. The reason for this phenomenon was that APP and MMT in APP + MMT mixture were separated in EP, while APP in APP-MMT complex was tightly mixed with MMT, which contributed to the charring reaction during combustion.

The flame-retardant performance of the nanocomposites mentioned in the article is shown in Table 5.

### 2.5. Metal-Containing Compounds

Metal-containing compounds are also one of commonly used flame retardants in practice applications. Especially, metal hydroxides or oxides are the most representative metal-containing compounds, which have the merit of low smoke, low toxicity and low cost [169,175]. Adding metal hydroxide to polymers can improve the flame retardancy and smoke suppression of polymers. Generally, they are inefficient when used alone, the added amount is 50%~70% of the flame-retardant materials to achieve the required flame-retardant level [176,177]. A large number of additions will result in significant reduced processing properties and mechanical properties of the composites. A group of researchers have found that materials containing Fe, Co or Ni elements can serve as catalysts to improve the reactivity of carbon deposition [178,179]. The combination of metal elements and intumescent flame retardants could inhibit the release of ammonia and carbon dioxide, and also suppress the generation of combustible gases [180,181].

In recent years, metal-containing flame retardants have aroused much interest. Q. Yin [182] et al. prepared metal-phosphorus hybrid nanomaterials by hydrothermal reaction, including aluminum phosphinate/phosphonate (APHNR/APHNSH), ferric aluminum phosphinate/phosphonate (FPHNR/FPHNSH), and zinc phosphinate/phosphonate (ZPHNR/ZPHNSH). It was found that metal hypophosphites and phosphates showed the morphology of nanorods and nanosheets. Among them, nanorods and nanosheets are formed by the accumulation of a multitude of polymer species generated by the reaction of metal substrates and phosphinic acid. The experimental results showed that blending EP/APHNR and EP/FPHNSH into EPs could obviously improve the fire resistance. The LOI values of EP/APHNR and EP/FPHNSH composites reached 29.8%. The reason for the improved flame-retardant performance was that the formation of stable P–O and Fe–O bonds by the pyrolysis reaction of metal centers and phosphorus with EP inhibited the release of flammable gases. Recently, it has been reported that metal-containing compounds were used as synergists to improve the flame-retardant efficiency of intumescent flame-retardant systems. A series of intumescent flame retardant EPs (IFR-EPs) by incorporating APP and metal compounds (Figure 19) was synthesized by Y.-Z. Wang [183] et al. The results exhibited that the flame retardancy of IFR-EPs was significantly improved. The LOI value is slightly improved when the total content of the added flame retardant is 5 wt.%. In addition, the experimental results showed that the introduction of metal compounds has catalytic effect on APP-containing EPs and a promotion effect on the generation of inert gases. Melamine polyphosphate (MPP) is one of the most effective halogen-free candidates. It is commonly used in combination with other materials to enhance its own or others’ flame-retardant effects. B. Schartel [184] et al. added three different types of melamine polymetallic phosphates MPAlP, MPZnP and MPMgP to EP, and compared their flame retardant properties. Moreover, using MgZnP in combination with other flame retardants (include MPP, diethyl aluminum phosphinate (AlPi-Et), 6H-dibenz[c,e][1,2]oxaphosphorin-6-propanoic acid, butyl ester, 6-oxide (DOPAc-Bu), boehmite (AlO(OH)), and SiO_2_) to research their synergistic flame retardant effect. The experimental results showed that all three flame retardants could impart flame retardancy to EP. MPAlP, MPZnP and MPMgP both worked in the gas phase and the condensed phase, but mainly by increasing the residue and protective layer in the condensed phase. The T_g_ of all composites exceeded 150 °C. The introduction of MPAlP halved pk-HRR, while MPZnP and MPMgP reduced pk-HRR by more than 70%. In addition, it was found that the combination of MPZnP and MPP had a synergistic effect on improving the flame retardancy.

The flame retardant performance of the metal-containing compounds mentioned in the article is shown in Table 6.

## 3. Summary and Perspectives

This paper covers the flame retardants of EP currently available or under serious development. At present, researches on the modification of EPs have made great progress, but with the development of technology, the requirements of EP composites have become higher. Currently, the performance of EP cannot meet the needs of high-tech fields. It is difficult for a single element to meet the requirements of efficient flame retardancy, and much work is still needed to improve the comprehensive performance of EPs. It is necessary to explore the use of new technologies and a combination of multiple synergistic flame-retardant technologies to modify EP.

The flame-retardant EP system is a complex system consisting of various components, including flame retardants, carbonization promoters, coupling agents and crosslinking agents, smoke suppressing agents, fillers and the like of different properties. Obviously, such a complex system cannot fully guide its formulation by a few empirical rules and general scientific principles. In order to obtain the optimized overall performance of this system, it is necessary to study the interaction between these components to exert the gain between them and avoid the harmful and mutually offsetting effects between them. Future research directions for EP flame retardants focus on environmentally friendly materials, sustainable materials, multifunctional additives and synergistic flame retardants.

## Figures and Tables

**Figure 1 materials-13-02145-f001:**
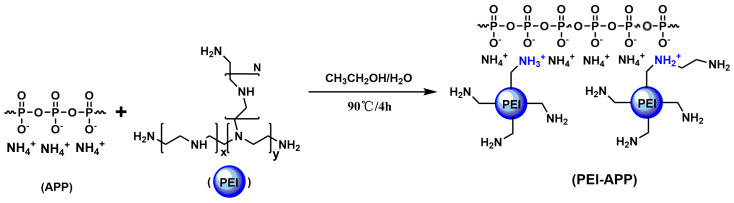
Synthesis route of polyethyleneimine (PEI-APP). Readapted with permission from Ref. [56], Copyright 2016, Elsevier Ltd.

**Figure 2 materials-13-02145-f002:**
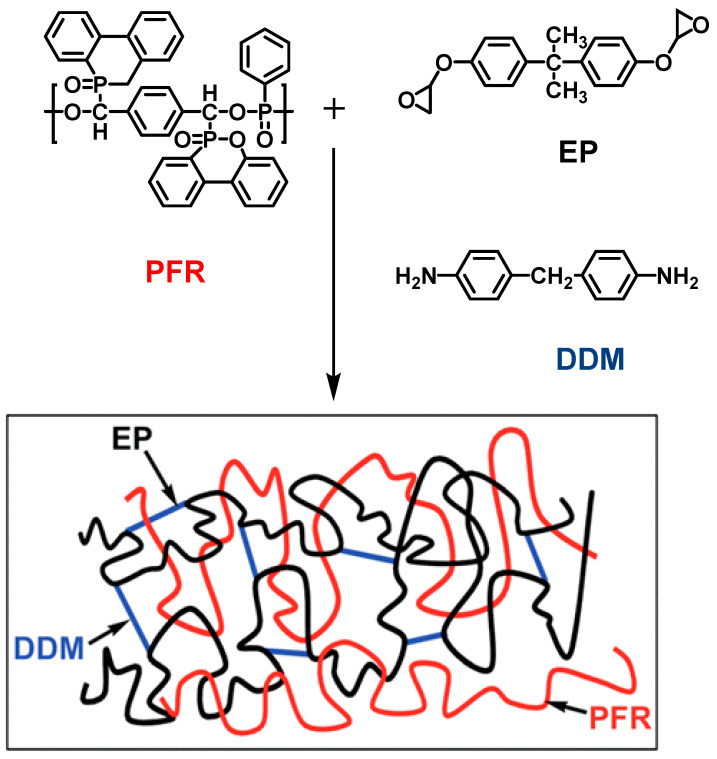
Preparation of polyphosphate flame retardant-epoxy resin (PFR-EP) semi-interpenetrating polymer network (SIPN) composites. Readapted with permission from Ref. [67], Copyright 2014, The Royal Society of Chemistry.

**Figure 3 materials-13-02145-f003:**
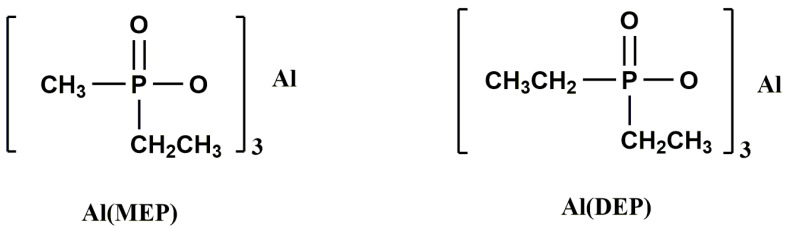
Structures for Al (MEP) and Al (DEP). Readapted with permission from Ref. [70], Copyright 2011, Society of Plastics Engineers.

**Figure 4 materials-13-02145-f004:**
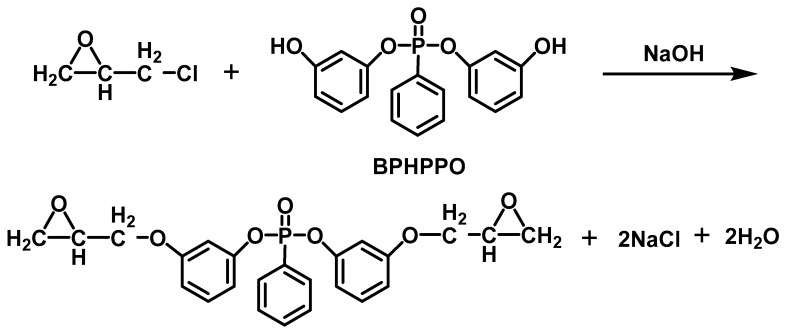
The synthesis of the BPHPPO-EP. Readapted with permission from Ref. [71], Copyright 2007, Elsevier Ltd.

**Figure 5 materials-13-02145-f005:**
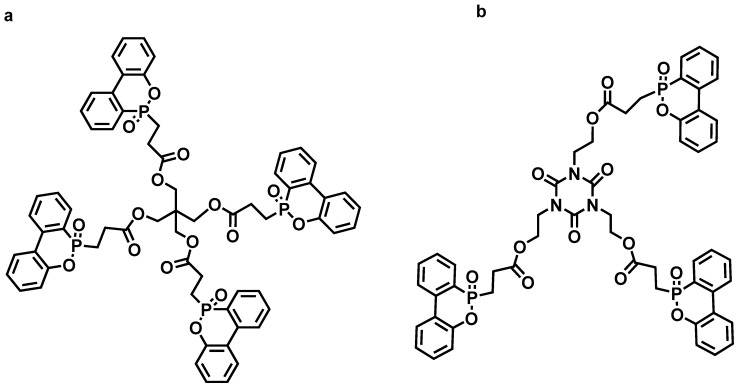
Structures of two phosphorus-containing flame retardants: (**a**) DOPP and (**b**) DOPI. Readapted with permission from Ref. [81], Copyright 2011, Elsevier Ltd.

**Figure 6 materials-13-02145-f006:**
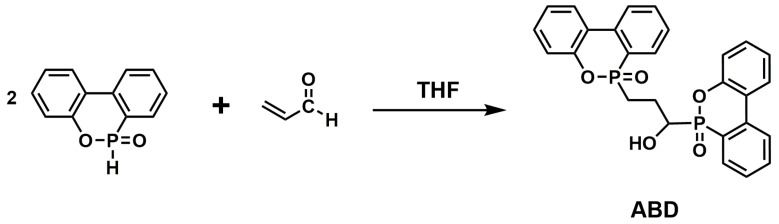
Synthesis route of ABD. Readapted with permission from Ref. [51], Copyright 2019, Elsevier Ltd.

**Figure 7 materials-13-02145-f007:**
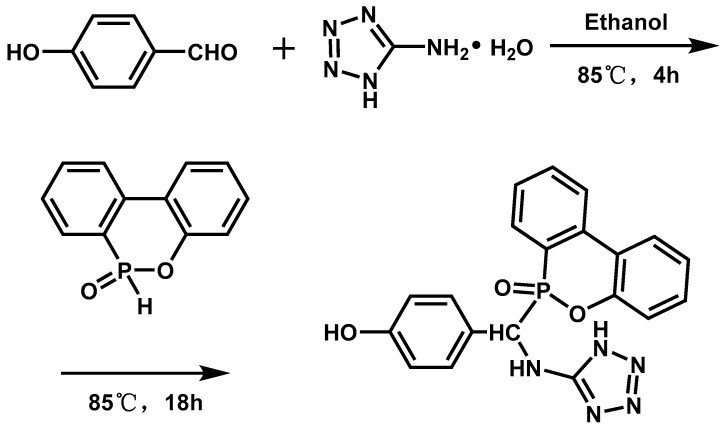
Synthesis route of DOPO-based tetrazole derivative (ATZ). Readapted with permission from Ref. [83], Copyright 2019, Elsevier B.V.

**Figure 8 materials-13-02145-f008:**
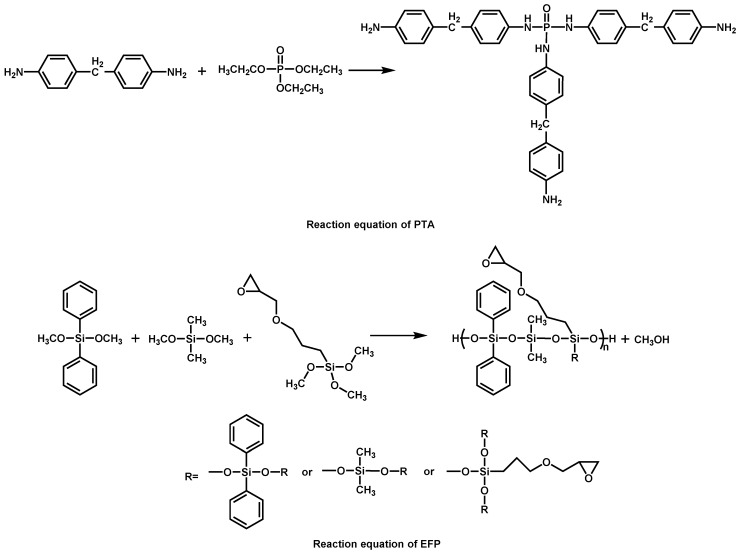
Reaction equation of phosphorous triamide (PTA) and epoxy-functional polysiloxane (EFP). Readapted with permission from Ref. [87], Copyright 2019, Elsevier Ltd.

**Figure 9 materials-13-02145-f009:**
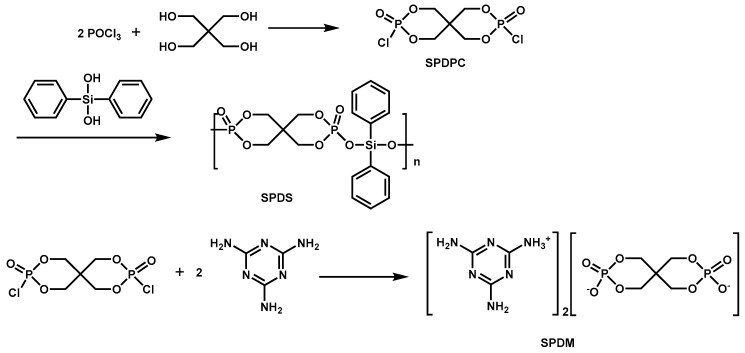
Synthetic routes to silicon/phosphorus-containing hybrid (SDPS) and mono (4, 6-diamino-1,3,5-triazin-2-aminium) mono (2,4,8,10-tetraoxa-3,9-di-phosphaspiro [5.5] undecane-3, 9-bis (olate) 3, 9-dioxide) (SPDM). Readapted with permission from Ref. [94], Copyright 2014, Elsevier Ltd.

**Figure 10 materials-13-02145-f010:**
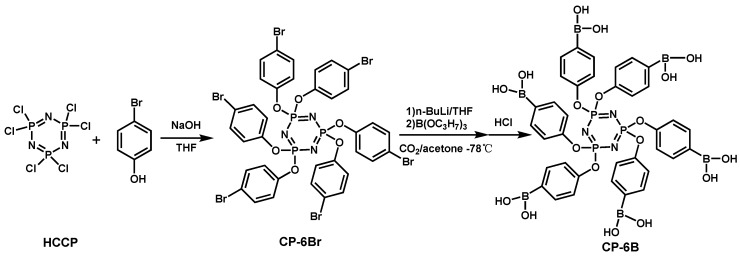
Synthesis route of CP-6B. Readapted with permission from Ref. [99], Copyright 2018 Elsevier Ltd.

**Figure 11 materials-13-02145-f011:**
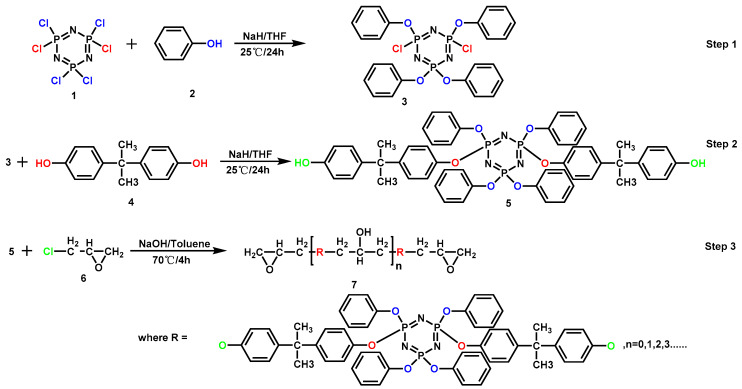
Synthesis route of cyclolinear cyclotriphosphazene-linked EP. Readapted with permission from Ref. [100], Copyright 2012, American Chemical Society.

**Figure 12 materials-13-02145-f012:**
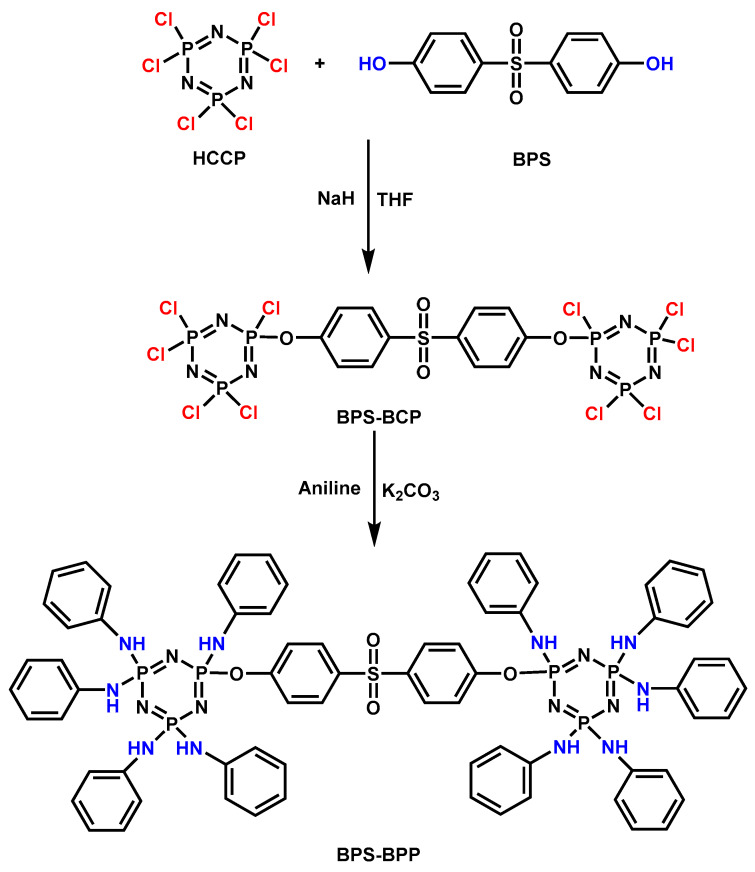
Synthetic route of BPS-BPP. Readapted with permission from Ref. [103], Copyright 2016 Elsevier Ltd.

**Figure 13 materials-13-02145-f013:**
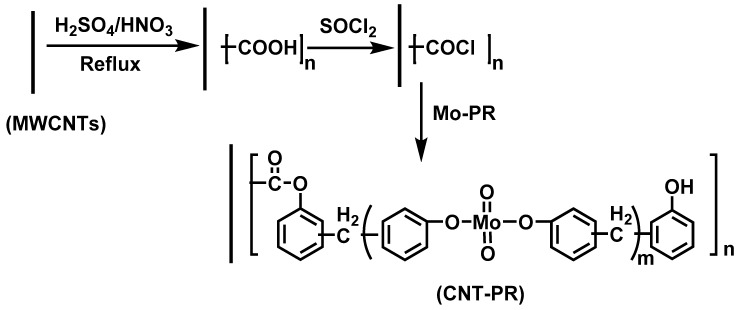
Reaction scheme of epoxy/CNT composites. Readapted with permission from Ref. [122], Copyright 2011, Elsevier Ltd.

**Figure 14 materials-13-02145-f014:**
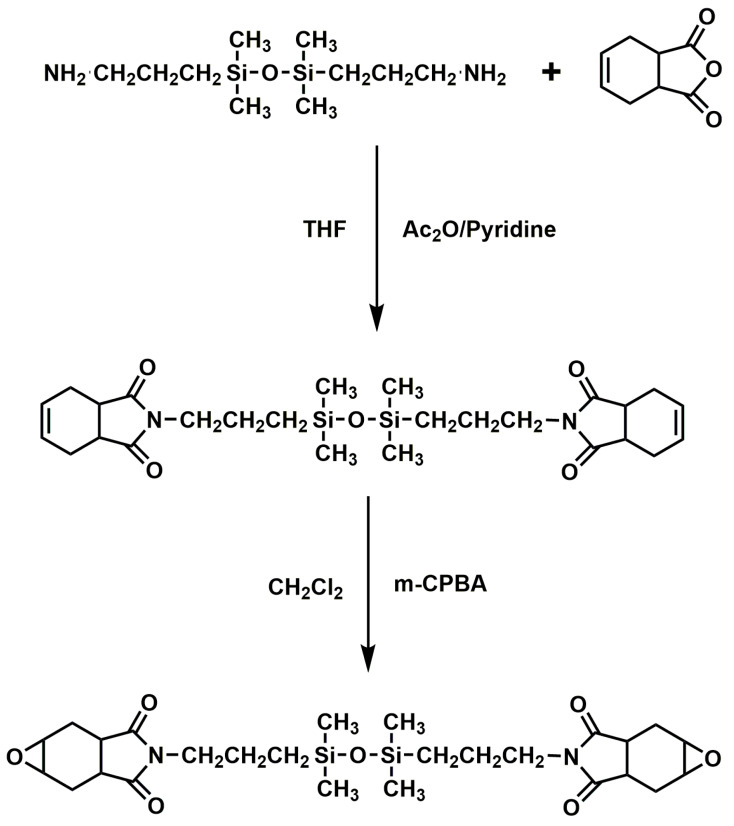
Synthesis of imide ring and siloxane-containing cycloaliphatic epoxy compound BISE. Readapted with permission from Ref. [142], Copyright 2007, Elsevier Ltd.

**Figure 15 materials-13-02145-f015:**
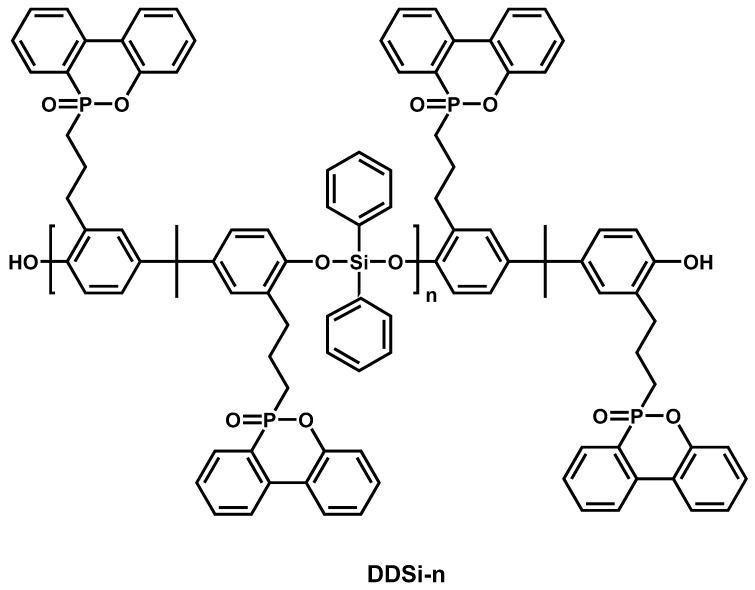
Chemical structure of the contrastive DDSi-n. Readapted with permission from Ref. [144], Copyright 2019, Elsevier Ltd.

**Figure 16 materials-13-02145-f016:**
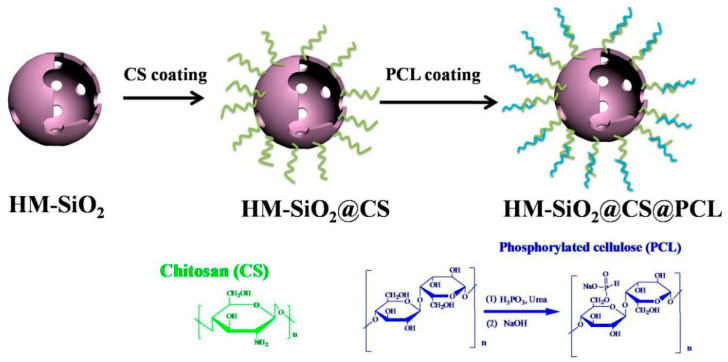
The synthetic route of HM-SiO2@CS@PCL. Readapted with permission from Ref. [149], Copyright 2017 Elsevier B.V.

**Figure 17 materials-13-02145-f017:**
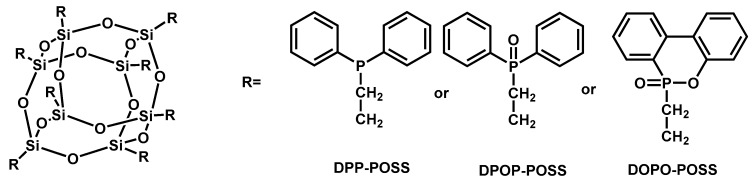
Structure of DPP-POSS, DPOP-POSS, and DOPO-POSS. Readapted with permission from Ref. [160], Copyright 2007, Elsevier Ltd.

**Figure 18 materials-13-02145-f018:**
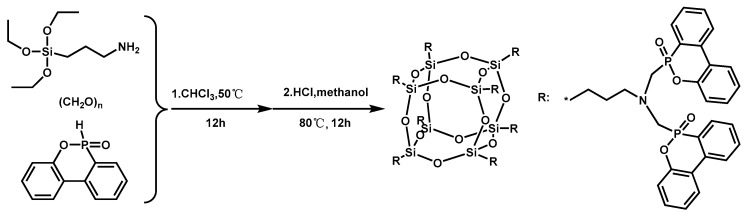
Synthetic route of ODMAS. Readapted with permission from Ref. [161], Copyright 2017, The Royal Society of Chemistry.

**Figure 19 materials-13-02145-f019:**
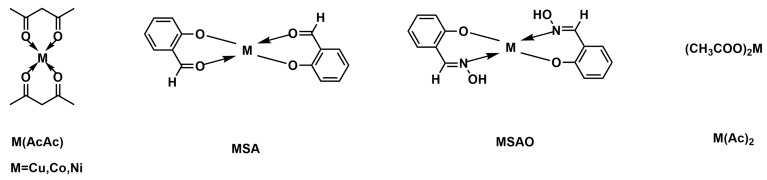
Structure of metal compounds. Readapted with permission from Ref. [183], Copyright 2009, Elsevier Ltd.

**Table 1 materials-13-02145-t001:** Limit oxygen index (LOI) and UL-94 data of epoxy thermosets at different formulation ratios. Readapted with permission from Ref. [59], Copyright 2015, American Chemical Society.

Sample	E51 (wt.%)	PA650 (wt.%)	MAPP (wt.%)	Cu_2_O (wt.%)	UL-94	LOI (%)
Neat EP	55.56	44.44			NR	19.0
EP/20%MAPP	44.44	35.56	20.00		NR	31.0
EP/20%Cu_2_O	44.44	35.56		20.00	NR	21.5
EP/16%MAPP/4%Cu_2_O	44.44	35.56	16.00	4.00	V-0	33.5
EP/18%MAPP/2%Cu_2_O	44.44	35.56	18.00	2.00	V-0	35.0
EP/18.67%MAPP/1.33%Cu_2_O	44.44	35.56	18.67	1.33	V-0	33.5
EP/19%MAPP/1%Cu_2_O	44.44	35.56	19.00	1.00	V-0	32.5
EP/13.5%MAPP/1.5%Cu_2_O	47.22	37.78	13.50	1.50	V-0	30.0
EP/9%MAPP/1%Cu_2_O	50.00	40.00	9.00	1.00	V-0	28.0
EP/8.1%MAPP/0.9%Cu_2_O	50.56	40.44	8.10	0.90	V-1	24.5

NR means no rating.

**Table 2 materials-13-02145-t002:** The flame retardancy performance of epoxy containing phosphorus flame retardants.

EPs and Incorporated Phosphorus Flame Retardant	wt.%	LOI (%)	UL-94	pk-HRR (kW/m^2^)	THR (MJ/m^2^)	Ref.
EP/PEI	1.2	23	NR	1074	45	[56]
EP/PEI-APP	10	26	V-1	280	16	[56]
	15	29.5	V-0	281	11	[56]
EP/4,4-diaminodiphenyl methan (DDM)	0(P)	24	–	–	–	[57]
EP/PA-I	4.16(P)	34	–	–	–	[57]
EP/PA-II	5.1(P)	35	–	–	–	[57]
EP/MAPP/Cu_2_O	0/0	19	NR	761	112	[59]
	20/0	31	NR	391	53	[59]
	18/2	35	V-0	312	61	[59]
EP/HPE	0	23	–	1250 ± 10	–	[63]
	33	27.5	–	491.8	–	[63]
	66	29.3	–	391 ± 5	–	[63]
	100	32	–	285 ± 5	–	[63]
EP/hb-FRs	0	18.7	HB	–	–	[64]
EP/hb-polyphosphoramide	10	23.3	HB	–	–	[64]
EP/hb-polyphosphordiamidate	10	22.6	HB	–	–	[64]
EP/hb-polyphosphoramidate	10	22.5	HB75	–	–	[64]
EP/hb-polyphosphate	10	22.1	HB	–	–	[64]
EP/ITA-HBP	0	26.4	NR	678.7	157.9	[66]
	3.8	36.4	V-0	618.6	135.7	[66]
	7.35	37.4	V-0	564.5	135.3	[66]
	10.64	41.6	V-0	534	125.9	[66]
	13.7	42	V-0	468	110.2	[66]
EP/PFR	0	26.4	NR	275.5	56.8	[67]
	7.36	34.3	V-1	231.7	64.8	[67]
	14.8	40	V-0	164.4	57.4	[67]
	22.28	42.2	V-0	150.8	46.4	[67]
EP/Al (MEP)	0	20.2	NR	–	–	[70]
	15	32.2	V-0	–	–	[70]
	20	36.4	V-0	–	–	[70]
EP/Al (DEP)	15	29.8	V-0	–	–	[70]
	20	35.5	V-0	–	–	[70]
EP/BPHPPO	0	22.5	–	–	–	[71]
	7.79(P)	34	–	–	–	[71]
EP	0	25	HB	1719	74.2	[81]
EP/DOPP	19.6	37.9	V-1	1191	44.8	[81]
EP/DOPI	23.1	34.2	V-0	869	41.5	[81]
EP-CF	0	33.2	HB	347	26.2	[81]
EP-CF/DOPP	5.9	45.3	V-0	248	19.9	[81]
EP-CF/DOPI	6.9	47.7	V-0	247	20	[81]
EP/ABD	0	24.7	NR	1420	143.6	[51]
	2	32.2	V-1	–	–	[51]
	3	36.2	V-0	1043	101.5	[51]
	4	39.1	V-0	933	94.3	[51]
EP/BDM	0	32.5	V-1	825	71	[82]
	5	34	V-0	812	68	[82]
	10	34.6	V-0	755	65	[82]
	20	35	V-0	683	60	[82]
	30	35.5	V-0	615	58	[82]
EP/ATZ	0	25.7	NR	654.3	100.3	[83]
	3	31.2	V-1	–	–	[83]
	6	33.7	V-0	482.5	83.9	[83]
	9	35.9	V-0	–	-	[83]
EP/DHBAP	0	25.8	NR	1063.1	76.1	[84]
	5	32.4	V-1	–	–	[84]
	8	34	V-0	783.3	59.9	[84]
EP/DHBAP	10	34.3	V-0	–	–	[84]
EP/DOPO-POSS	0	25	NR	855	112	[92]
	1.5	29	V-1	–	–	[92]
	2.5	30.2	V-1	969	103	[92]
	3.5	29.1	V-1	–	–	[92]
	5	28.5	NR	588	92	[92]
	10	23	NR	483	85	[92]
EP/ZrPP/POSS	0/0	23	NR	675	–	[93]
	0/5	27.6	V-2	426	–	[93]
	1/4	30.3	V-1	438	–	[93]
	3/2	29.7	V-1	461	–	[93]
	5/0	28.4	V-2	469	–	[93]
EP/PTA/EFP	0/0	20.7	NR	–	–	[87]
	10/0	23.5	NR	–	–	[87]
	0/10	24.8	NR	–	–	[87]
	5/5	30.2	V-1	–	–	[87]
EP/SPDS/SPDM	0/0	20.2	–	1650	213	[94]
	5.5/0	28.9	–	1378	203	[94]
	0/5.5	25.1	–	–	–	[94]
	5.2/5.2	30.8	–	1122	207	[94]
EP/CP-6B	0	22.8	NR	1026	83	[99]
	1	29.6	V-1	709	78	[99]
	3	30.8	V-0	599	74	[99]
	5	31.4	V-0	446	58	[99]
	7	32.3	V-0	359	54	[99]
	10	28.6	V-0	471	60	[99]
CL-CTPN-EP/Dicyandiamide	–	32.4	V-0	–	–	[100]
CL-CTPN-EP/DDM	–	31.6	V-0	–	–	[100]
CL-CTPN-EP/novolak	–	30.2	V-0	–	–	[100]
EP/BPS-BPP	0	21.5	NR	1000	89	[103]
	3	27.5	NR	–	–	[103]
	6	28.7	V-1	–	–	[103]
	9	29.7	V-1	537	76	[103]
	12	28.3	V-1	–	–	[103]
EP/CTP-DOPO	0	21.7	NR	619.8	77.6	[104]
	9.7	34.3	V-1	–	–	[104]
	10.6	36.6	V-0	282.6	51.7	[104]
	11.7	38.5	V-0	–	–	[104]
	12.6	39.8	V-0	–	–	[104]
EP/HPCTP/OGPOSS	0/0	–	NR	1321	157	[105]
	15/0	–	V-0	1026	145	[105]
	10/5	–	V-0	707	123	[105]
	7.5/7.5	–	V-0	581	110	[105]
	5/10	–	V-0	560	105	[105]
	0/15	–	NR	513	82	[105]

“NR” means no rating, “–” means no test data.

**Table 3 materials-13-02145-t003:** The flame retardancy performance of epoxy containing carbon-based materials.

EPs and Incorporated Carbon-Based Materials	wt.%	LOI (%)	UL-94	pk-HRR (kW/m^2^)	THR (MJ/m^2^)	Ref.
EP	2	–	–	1653	129.9	[114]
EP/GNS	2	–	–	1156	107.8	[114]
EP/Ce-MnO_2_	2	–	–	920	96.7	[114]
EP/Ce-MnO_2_-GNS	2	–	–	765	83.8	[114]
EP	2	–	–	1348	87.1	[115]
EP/MoS_2_	2	–	–	1076	75.7	[115]
EP/GNS	2	–	–	965	70.1	[115]
EP/MoS_2_/GNS	2	–	–	730	65.1	[115]
EP	2	–	–	1730	113.1	[116]
EP/GNS	2	–	–	980	65.1	[116]
EP/Ni-Fe LDH	2	–	–	1070	58.9	[116]
EP/Ni-Fe LDH/GNS	2	–	–	678	44.2	[116]
CF-EP	0	–	–	568	23.2	[120]
CF-EP/SWCNT-buckypaper	1	–	–	526	24.5	[120]
CF-EP/MWCNT-buckypaper	1.3	–	–	258	13.2	[120]
CF-EP/CNF paper	1.5	–	–	508	24.8	[120]
EP/VETS-CNT	0	22	V-1	–	–	[121]
	1	23	V-1	–	–	[121]
	3	25	V-0	–	–	[121]
	5	26	V-0	–	–	[121]
	7	27	V-0	–	–	[121]
	9	29	V-0	–	–	[121]
EP	0	21.5	NR	900	–	[122]
EP/Melamine	8	22.4	NR	750	–	[122]
EP/Mo-PR/Melamine	2/0	24	NR	543	–	[122]
	1/8	28	V-2	–	–	[122]
	2/8	29.5	V-0	579	–	[122]
EP/CNT-PR/Melamine	1/8	27.7	V-2	527	–	[122]
	3/8	28.6	V-2	535	–	[122]
	5/8	29.5	V-0	468	–	[122]
EP/CF/CNT	0/0	–	–	971.7	98.8	[123]
	0.5/0	–	–	792.7	92.5	[123]
	0.7/0	–	–	722.6	88.2	[123]
	1/0	–	–	840.2	88.9	[123]
	1.5/0	–	–	793.3	101.7	[123]
	0.5/0.5	–	–	648.1	75	[123]
	0.7/0.7	–	–	635	80.3	[123]
	1/0.5	–	–	701.7	99.3	[123]
EP/PPMS-CNT	0	19.3	HB	1334.6	100.1	[124]
	5	21.5	HB	1013.4	93.7	[124]
	10	22.6	V-2	680.7	90.7	[124]
	15	24.5	V-2	444.6	77.6	[124]
epoxy- aliphatic amine system/EG	0	18.7	–	–	–	[130]
	1	21.3	–	–	–	[130]
	2	25	–	–	–	[130]
	3	28	–	–	–	[130]
	4	30	–	–	–	[130]
Jatropha curcas oil-based alkyd-EP/EG	0	18	–	–	–	[131]
	0.5	21	–	–	–	[131]
	1.5	24	–	–	–	[131]
	2.5	29	–	–	–	[131]
	4	35	–	–	–	[131]
	5	41	–	–	–	[131]

“NR” means no rating, “–” means no test data.

**Table 4 materials-13-02145-t004:** The flame retardancy performance of epoxy containing silicon flame retardants.

EPs and Incorporated Silicon Flame Retardants	wt.%	LOI (%)	UL-94	pk-HRR (kW/m^2^)	THR (MJ/m^2^)	Ref.
EP	0	26.4	NR	1420	144	[144]
EP/DDSi-1	4	32.4	V-1	1115	105	[144]
	6	34.1	V-1	907	101	[144]
	8	35.9	V-1	743	95	[144]
EP/DDSi-2	8	34.8	V-0	779	98	[144]
EP/DDSi-5	8	33	V-0	892	95	[144]
EP	0	26.4	NR	1420	144	[145]
EP/DDSi-1	4	32.4	V-1	1115	105	[145]
EP/TriDSi	4	33.4	V-1	–	–	[145]
EP/TetraDSi	4	34.6	V-1	–	–	[145]
EP/DDSi-1	6	34.1	V-1	907	101	[145]
EP/TriDSi	6	35.2	V-0	810	90	[145]
EP/TetraDSi	6	36	V-0	776	83	[145]
EP	0	–	–	1377.7	86.6	[149]
EP/HM-SiO_2_	0.5	–	–	1226.4	67	[149]
	2	–	–	860.6	69.8	[149]
EP/HM-SiO_2_@CS@PCL	0.5	–	–	791.8	67.2	[149]
	2	–	–	676.3	86.3	[149]
EP	0	20.6	NR	811.1	114.2	[152]
EP/RH-SiO_2_/DOPO-J-ESO	20/0	30.9	NR	520	78.8	[152]
	20/5	33.2	V-0	482.2	52.9	[152]
	20/10	35.8	V-0	436.8	41.5	[152]
	20/15	36.9	V-0	425.9	36	[152]
EP /DOPO-J-ESO	10	31.8	V-0	506.2	71.5	[152]
EP	0	23	NR	893	112	[160]
EP/DPP-POSS	5	33.2	V-0	489	94.1	[160]
EP/DPOP-POSS	5	29.3	V-1	419	87.8	[160]
EP/DOPO-POSS	5	30	V-1	433	91.1	[160]
EP/ODMAS	0	25.6	NR	–	–	[161]
	1	29.7	V-1	–	–	[161]
	5	35.5	V-0	–	–	[161]
	10	36.5	V-0	–	–	[161]
	15	37.1	V-0	–	–	[161]
EP/POSS-bisDOPO	0	25.4	–	–	–	[162]
	1	29.3	–	–	–	[162]
	5	31.7	–	–	–	[162]
	10	33.2	–	–	–	[162]
	20	34.5	–	–	–	[162]
EP/DOPO	5	29.7	–	–	–	[162]
EP/POSS	5	26.9	–	–	–	[162]
EP/DOPO + POSS	5	29.2	–	–	–	[162]

“NR” means no rating, “–” means no test data.

**Table 5 materials-13-02145-t005:** The flame retardancy performance of epoxy containing nanocomposites.

EPs and Incorporated Nanocomposites	wt.%	LOI (%)	UL-94	pk-HRR (kW/m^2^)	THR (MJ/m^2^)	Ref.
EP	0	21.8	NR	781	107	[172]
EP/DiDOPO	1	24.1	V-2	–	–	[172]
	3	32.7	V-0	–	–	[172]
	7	35.7	V-0	491	80	[172]
EP/OMMT	1	22.4	NR	–	–	[172]
	3	23.7	NR	–	–	[172]
	7	23.7	NR	576	98	[172]
EP/DiDOPO/OMMT	0.5/0.5	23.2	NR	–	–	[172]
	1.5/1.5	27.1	V-0	–	–	[172]
	3.5/3.5	32.2	V-0	369	95	[172]
EP	0	21	–	–	–	[173]
EP/MMT/MWCNT	1.5/0.1	26.4	–	–	–	[173]
EP/Fluorinated MMT	1.5	25.8	–	–	–	[173]
EP/Fluorinated MWCNT	0.1	23	–	–	–	[173]
EP/Fluorinated MMT/Fluorinated MWMMT	1.5/0.1	31	–	–	–	[173]
EP	0	23	NR	860	112	[174]
EP/APP	10	25	NR	458	62	[174]
EP/APP + MMT	10	28	V-0	524	50	[174]
EP/APP-MMT	10	30	V-0	393	34	[174]

“NR” means no rating, “–” means no test data.

**Table 6 materials-13-02145-t006:** The flame retardancy performance of epoxy containing metal-containing compounds.

EPs and Incorporated Metal-Containing Compounds	wt.%	LOI (%)	UL-94	pk-HRR (kW/m^2^)	THR (MJ/m^2^)	Ref.
EP	0	22.2	–	–	–	[182]
EP/APHNR	7.8	29.8	–	–	–	[182]
EP/APHNSH	7.8	26.4	–	–	–	[182]
EP/FPHNR	7.8	27.4	–	–	–	[182]
EP/FPHNSH	7.8	29.8	–	–	–	[182]
EP/ZPHNR	7.8	22.2	–	–	–	[182]
EP/ZPHNSH	7.8	25.2	–	–	–	[182]
EP	0	19.6	NR	939	179	[183]
EP/APP	5	27.1	V-0	283	111	[183]
EP/APP-CoSA	4.97/0.03	28	V-0	–	–	[183]
	4.92/0.08	28.4	V-0	–	–	[183]
	4.83/0.17	29.4	V-0	310	95	[183]
	4.75/0.25	28.4	V-0	–	–	[183]
	4.67/0.33	28.7	V-0	–	–	[183]
EP	0	–	HB	1068	75.8	[184]
EP/MPAlP	20	–	HB	540	60	[184]
EP/MPZnP	20	–	HB	312	60	[184]
EP/MPMgP	20	–	V-1	298	57.3	[184]
EP/MPZnP + MPP	10/10	–	V-1	207	51.1	[184]
	6.7/13.3	–	V-0	211	32.5	[184]
EP/MPZnP + AlPi-Et	10/10	–	HB	405	51.2	[184]
	6.7/13.3	–	V-1	435	53.8	[184]
EP/MPZnP + DOPAc-Bu	10/10	–	V-1	329	57.6	[184]
	6.7/13.3	–	HB	412	52.1	[184]
EP/MPZnP + AlO (OH)	10/10	–	HB	438	57.2	[184]
	6.7/13.3	–	HB	575	57.9	[184]
EP/MPZnP + SiO_2_	10/10	–	HB	525	62.4	[184]
	6.7/13.3	–	HB	681	65.6	[184]
EP/MPP	20	–	V-0	244	26.6	[184]
EP/AlPi-Et	20	–	V-0	492	55.8	[184]
EP/DOPAc-Bu	20	–	HB	624	50.2	[184]
EP/AlO (OH)	20	–	HB	870	65.5	[184]
EP/SiO_2_	20	–	HB	907	57.6	[184]

“NR” means no rating, “–” means no test data.

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
