# Peer review of "Recent Developments in the Flame-Retardant System of Epoxy Resin"

_materials, 2020, doi:10.3390/ma13092145_

Round 1

Reviewer 1 Report

The authors have addressed all of my concerns from the previous review of the paper. 

Author Response

Your comments are very helpful to my manuscript. Thank you for reviewing my manuscript.

Reviewer 2 Report

Although the revised and resubmitted manuscript “Recent developments in the flame-retardant system of epoxy resin” has improved significantly its quality, some typographical errors (or formal aspects) should be corrected before its publication:

  • Line 77: Substitute “288 degrees Celsius” by “288 °C”.
  • Line 121: add % after LOI’ value (revise the rest of the manuscript).
  • Line 138: The weight percentage of MAPP and Cu2O should be identified with wt.% instead of %. The authors should check the rest of the manuscript to confirm that there are not similar typographic errors.
  • Line 148: Substitute “Eps” by EPs”.
  • Line 163: Separate “retardants” from “(HB-FRs)”. The authors should also revise all the manuscript to confirm that there are not identical errors in the manuscript.

Author Response

Point 1: Although the revised and resubmitted manuscript “Recent developments in the flame-retardant system of epoxy resin” has improved significantly its quality, some typographical errors (or formal aspects) should be corrected before its publication:

Line 77: Substitute “288 degrees Celsius” by “288 °C”.

Line 121: add % after LOI’ value (revise the rest of the manuscript).

Line 138: The weight percentage of MAPP and Cu2O should be identified with wt.% instead of %. The authors should check the rest of the manuscript to confirm that there are not similar typographic errors.

Line 148: Substitute “Eps” by EPs”.

Line 163: Separate “retardants” from “(HB-FRs)”. The authors should also revise all the manuscript to confirm that there are not identical errors in the manuscript.

Response 1: Thank you for taking the time to review my manuscript. I have corrected the errors in my manuscript based on your comments, and checked for other similar errors in the manuscript.

Reviewer 3 Report

The authors reported recent advances in the flame retardancy of epoxy resins. The resubmitted manuscript has been significantly enriched with a large number of important publications in this field when compared with the previous version. I recommend this article for publication in Materials, however, some corrections have to be made prior to acceptance. 

  1. The authors use abbreviations but do not explain their meaning (for example line 78 ATH, line 84 PWB...)
  2. Fig. 4. There are mistakes in chemical structures
  3. line 289 "polyhedral oligosiloxane (POSS)" it is not correct.
  4. Fig. 8. mistakes in chemical structures
  5. diphenylhydroxysilane is not correct compound name, it should be diphenylsilanediol
  6. again mistake in Fig. 9.
  7. a mistake in Fig. 12.
  8. Paragraph 2.3.1. Siloxane - there are two references 146 and 147, however, the chemical structures of the applied silicon-based flame retardants do not contain siloxane units. The siloxanes are compounds with the Si-O-Si linkage, not O-Si-O as presented in Fig. 18. That means that the authors of papers (Ref. 146, 147) also made mistakes in the classification of synthesized compounds.
  9. Paragraph Silica: "M. Gao [152] et al. synthesized a new type of
    polymer silicon-containing intumescent flame retardant (Si-IFR) from silica (Fig 19)." Comment: The Si-IFR has been synthesized from chlorosilane, not from silica. That means that the location of this reference is not correct. 
  10. Fig. 20. a mistake in chemical structure.
  11. The manuscript contains many editorial mistakes and should be carefully checked.  

Reviewer 4 Report

This is an extensive review of the use of non-halogenated flame retardants (FRs) used to help protect epoxy resins (EPs). There are 188 references cited, many of them published in the past few years. The authors have reviewed a wide range of different classes of FRs, and summarize the findings of the previous works cited. The summary, while concise, is accurate and helpful. As such, this is an important publication that will be of considerable value to both basic researchers and engineers in this field. I recommend publication with some changes as suggested below. The changes do not significantly change the focus of the review, but they could make the manuscript more valuable and shorter.

  1. It is properly stated in the review about the health and environmental concerns of some of the chemicals used as FRs, especially the halogenated species. However, many, if not most of the replacements for halogenated FRs may not have sufficient testing and data to show if they are indeed safer to humans and the environment. For example, there is a recent review article discussing organophosphate FRs (OPFRs). As the title suggest, those authors (as well as many others) consider OPFRs as a class of emerging contaminants (A Review of a Class of Emerging Contaminants: The Classification, Distribution, Intensity of Consumption, Synthesis Routes, Environmental Effects and Expectation of Pollution Abatement to Organophosphate Flame Retardants (OPFRs). Jiawen Yang, Yuanyuan Zhao, Minghao Li, Meijin Du, Xixi Li, and Yu Li. Int J Mol Sci. 2019 Jun; 20(12): 2874. Published online 2019 Jun 12. doi: 10.3390/ijms20122874. PMCID: PMC6627825, PMID: 31212857).

I realize that it is almost impossible to do a comprehensive review of the methods and materials used to improve fire safety while also performing a comprehensive review of their associated health effects. I would suggest that a paragraph of two be added early in the paper to discuss how new FR chemicals may be a pollution issue, and that additional research on this is needed in addition to research on the effectiveness of the materials as an FR. It is important for FR researchers to know that the history of FR chemicals is filled with regrettable substitutions, in part because the long-term health effects may not become apparent for years or decades after new FR chemical are introduced into the market.

  1. I think that many of the figures are not necessary. This review is not about how the new materials are produced, or the reaction mechanisms involved. I would consider removing figures such as 2 (bottom half), 13, 14, 16, 22, 23, and 24. The other figures should be reviewed to determine if they are actually adding sufficient information to warrant their inclusion.
  2. There are many values presented in the text, including such factors as concentrations, heat release rates, which flammability standard was passed, and issues such as mechanical properties being degraded. It might be better to include a table for each class of FRs that would try to summarize these numbers, allowing for a reader to more quickly scan the results. Items could include positive benefits as well as highlighting potential issues. Reducing the number of figures would allow for more tables (there is only one table).
  3. I would revise the text to remove subjective statements such as “excellent” (23 instances) or “wonderful” (2 instances) when referring to certain properties. I am not sure how a substance is determined to have “excellent” properties as no criteria is given (and I don’t believe they exist). The quantitative values presented are sufficient. I would note that these words do appear in some of the titles of cited works.
  4. A similar suggestion to #4 is to remove speculative statements that appear in several places. These include “the authors believe” (line 630) or “the authors considered” (line 686) (meaning the authors of the original papers, not the authors of the review paper) when referring to how the FR works. Unless there is experimental evidence these statements do not add much.
  5. I am not sure why certain sections appear as red text. It is not clear that all of the red text is of significantly higher importance than other sections.

Round 2

Reviewer 3 Report

The authors have corrected all mistakes in the manuscript, which I listed in the review, therefore, I would like to recommend this manuscript for publication in Materials.

This manuscript is a resubmission of an earlier submission. The following is a list of the peer review reports and author responses from that submission.

Round 1

Reviewer 1 Report

I want to commend the authors on compiling a long list of epoxy flame retardant papers.  However, this review paper is effectively just a long list of peer reviewed flame retardant papers, with no review of commercial technologies, past reviews on the subject, or analysis of the results and the strengths/weaknesses of each of the different chemistries. For reviews to be meaningful, they need to not just summarize the results of the papers cited, but also provide meaning about how the flame retardant chemistry did/did not provide flame retardancy vs. other materials.  For example - the flame retardant chemistry could be discussed in regards to vapor phase vs. condensed phase mechanisms, unexpected (or expected) synergistic/antagonistic fire behavior, and specific aspects of fire safety provided by a chemical class.  Was heat release reduced but at the negative expense of flame spread / dripping in a vertical burn test?   Was smoke decreased as heat release decreased?   Did the enhanced char formation yield any other fire safety benefits?   Without analysis, we have a long citation list, not a review paper.

There are some other minor points which need to be addressed in the paper as well.

1) Regarding reference #30 (page 1, lines 42-44),  yes, halogenated FRs do produce dioxins, but all fires produce toxic products. The main reason for scrutiny with halogen is not the fire issue, it's the negative aspects of persistence, bioaccumulation, and toxicity in the environment when the small molecule gets out of the epoxy.  This is also a problem for non-halogenated flame retardants as well if they are not covalently bonded to the epoxy. 

2) When referring to broader classes of flame retardants used today (example, sections 2.1.1 and 2.1.2) there are much more comprehensive reviews available which should be cited.  The reviews written by Levchik and Weil on flame retardants for epoxy, as well as books available on flame retardants in general, are much better sources of information than the articles the authors have chosen to cite in this paper. 

3) Section 2.2.3 - the word "Leprosy Fruit" is used in this section.  Please check if this is the correct name of the fruit in question - I went looking for it and I'm not sure it is the right word as I couldn't find any existence of a fruit by this name.  I found fruits that had been historically used to treat leprosy, but not a fruit of this name. 

4) Page 16, line 554:  This paper does not cover flame retardants currently available since no commercial flame retardants are discussed in this review.  Further, just because work is published does not mean it is under development.  Sometimes research does not progress beyond the initial lab study.  Again, analysis and context on what flame retardant chemistry in this paper shows great promise and should be developed, or is building off of known commercial flame retardant chemistry (such as DOPO) should be provided in this paper and toward the end of the paper to help the reader put the entire paper into context about the strength (or weakness) of the non-halogenated flame retardant offerings for epoxy. 

Reviewer 2 Report

The present work briefly reviews some advances on the development of flame retardants used to improve flame retardancy of epoxy resins.

Although, the way that the review is organized and presented is easy to read and well discussed the main achievements, more recent works (last 1-2 years) regarding with the use of flame retardants in EP should be included and discussed along the manuscript. For instance, lines 218-218, the authors mentioned that “Currently, ….phosphorus-silicon …of EPs. R. Yang [87]…”, nevertheless, the reference [87] is from 2011.

Furthermore, since a review regarding with the use of flame retardants to improve epoxy resins flame retardancy (Molecules 2019, 24(21), 3964, doi.org/10.3390/molecules24213964) has already been recently published, the authors should properly justify the innovation of the present work respect to other similar previously published.

Moreover, some typographic errors or formal aspects found in the manuscript should be corrected, such as:

- Addition of a space character before references’ numeration, such as: -  line 30: “thermal resistance[1-6]…”

- Adding a point at the end of all “et al.” abbreviations.

- Line 132: “… and used as the as curing agent…”

- Line 135: replace “total exothermic (THR)” by “total heat evolved (THE)”

- Replace “scholars” by researchers or authors. For instance: line 350 replace “…concerned by many scholars [112,113].” By “…concerned by many authors [112,113].”

- Lines 379-380: Remove sentence “There are numerous researches …. retardants [119,120].”

- Lines 450-451: Remove sentence: “Now, POSS….hybrids”.

- Lines 559-560: Remove subjective sentence: “It is believed…. fascinating materials”

Reviewer 3 Report

The authors reported recent advances in the flame-retardancy of epoxy resins. The topic is very important and all review articles are very valuable. Therefore, I recommend this article for publication, however, a major revision of the manuscript is required prior to acceptance.

  • First of all, the manuscript should be significantly improved by the addition of a greater number of chemical structures of described flame-retardants. It will be more eye-catchy for readers. Please see how other authors present their review articles (Journal of Applied Polymer Science: 10.1002/app.47910). Moreover, the chemical structures should be uniformed in the text, each chemical structure is exported from the different template and it looks very poor. 
  • The authors mentioned the hyperbranched P-based flame retardant, however, very important papers were omitted such as:
    Polymer Chemistry 2019 DOI: 10.1039/c9py00737g
    Chemical Engineering Journal DOI: 10.1016/j.cej.2019.122719
    European Polymer Journal DOI: 10.1016/j.eurpolymj.2019.109390

  • I found a mistake in the chemical structure given in 5b.

  • Paragraph 2.1.3. DOPO contains only several references, I believe that the authors will find and described more papers, due to the increasing interest of DOPO derivatives.
  • Moreover, the authors should add another paragraph and describe cyclophosphazenes, which are one of the most important classes of the inorganic compounds, which can be functionalized with amines, alcohols, phenols and other groups of compounds (organoboron DOI: 10.1016/j.polymdegradstab.2018.07.026, organosilicon DOI: 10.1016/j.jorganchem.2017.10.030; 10.1039/C8NJ03800G). It means, that the inorganic P=N core can survive under special conditions and also provides high thermal stability. The special properties of phosphazenes resulted in the application of these groups of compounds in many fields, including halogen-free flame retardants. I think that the phosphazenes deserve a separate paragraph. Exemplary references are given below, however, the literature survey should be done by the authors of this paper:

DOI: 10.1016/j.polymdegradstab.2018.07.026

DOI: 10.1021/ie400483x

DOI: 10.1021/ie300962a

DOI: 10.1039/c2ra20739g

DOI: 10.1016/j.polymdegradstab.2016.11.023

DOI: 10.1016/j.polymdegradstab.2015.11.018

DOI: 10.1177/0954008314528227 <- this reference also includes POSS compound

  • "Section 2.3.2. POSS" the meaning of the abbreviation should be given (polyhedral oligomeric silsesquioxanes). Moreover, several words about POSS compounds should be added to introduce the reader to this family of hybrid compounds (diameter, general formula, properties ref: C. Hartmann-Thompson, Applications of Polyhedral Oligomeric Silsesquioxanes, London-New York, 2011.; D. B. Cordes, P. D. Lickiss and F. Rataboul, Chem. Rev.,
    2010, 110, 2081–2173.). The growing interest of the application of POSS compounds is a result of easy modification with a wide range of reactions (hydrosilylation: 10.1002/cctc.201801609, nucleophilic substitution: 10.1055/s-0029-1216807 thiol addition:  10.1039/C9NJ04488D, P-H addition: 10.1016/j.compscitech.2016.02.026 and many more...)

omitted references about flame retardancy of POSS compounds in the epoxy resin:

Ref:

DOI: 10.1039/C5TA07115A

DOI: 10.1016/j.tca.2008.03.020 

DOI: 10.1016/j.polymdegradstab.2014.07.023

DOI: 10.1016/j.compscitech.2016.02.026

DOI: 10.1016/j.porgcoat.2009.04.008

  • In the introduction, authors postulate that "It is necessary to ensure that the other properties of the EP are not sacrificed while improving the flame-retardant performance" I fully agree with this sentence. I think that in the cited references many important measurements were done and it is worthy to mention the influence of the flame-retardants on the thermal, mechanical properties of resulting materials.